# FLEMO$_{flash}$ - Flood Loss Estimation MOdels for companies and households affected by flash floods

Apoorva Singh[1,2#] , Ravi Kumar Guntu[1#*] , Nivedita Sairam[1] , Kasra Rafiezadeh Shahi[1] , Anna Buch[3] , Melanie Fischer[1] , Chandrika Thulaseedharan Dhanya[2] , and Heidi Kreibich[1]

[1]Section 4.4 Hydrology, GFZ Helmholtz Centre for Geosciences, 14473 Potsdam, Germany

[2]Department of Civil Engineering, Indian Institute of Technology Delhi, 110016 New Delhi, India

[3]Institute of Geography, University of Heidelberg, 69117 Heidelberg, Germany

*Correspondence to*: Ravi Kumar Guntu (ravikumar.guntu@gfz.de)

#These authors contributed equally to this work

**Key points:**

- Applying Machine Learning on multi-event data reveals key drivers of flash flood losses such as flow velocity and emergency response.

- The first probabilistic flash flood loss model provides robust estimates of company and household losses, including uncertainty information.

- High preparedness during extreme flash floods was found to reduce building losses of households by up to 77%.

**Abstract.** In light of the increasing losses from flash floods intensified by climate change, there is a critical need for improved loss models. Loss assessments predominantly focus on fluvial flood processes, leaving a significant gap in understanding the key drivers of flash floods and the effect of preparedness on losses. To address these gaps, we introduce FLEMO$_{flash}$—a novel multivariate probabilistic Flood Loss Estimation Model compilation for flash floods. The models are developed for companies and households based on survey data collected after flash flood events in 2002, 2016, and 2021 in Germany. FLEMO$_{flash}$ employs a data-driven feature selection approach, combining machine learning techniques (Elastic Net, Random Forest, XGBoost) to identify key drivers influencing flash flood losses and Bayesian networks to model probabilistic loss estimates, including uncertainty. Model-based findings show that in extreme hazard scenarios, high preparedness can reduce building losses by up to 47% for large companies. Households who knew exactly what to do during high water depth were able to reduce their building losses by 77% and contents losses by 55%. Thus, FLEMO$_{flash}$ can support risk communication and management by providing reliable estimation of flash flood losses along with the loss differential considering the level of risk preparedness.

# 1 Introduction

Flash floods characterized by their rapid onset and short duration, rank amongst the most devastating natural disasters, leading to significant loss of life and property (Zain et al., 2021). The flash flood events in western Germany and neighbouring countries during July 2021 caused an estimated USD 54 billion in losses (Munich Re, 2025), marking it as the costliest natural

disaster in the history of Germany to date. Other notable examples include events in the Elbe and Danube rivers in Germany in 2002, which resulted in USD 9 billion losses (Kreibich et al., 2007). The 2017 flash floods in the Houston area of Texas during Hurricane Harvey resulted in losses ranging from USD 90 to 160 billion (Rözer et al., 2021). The 2012 event in Beijing caused total losses of USD 1.86 billion (Wang et al., 2013), and the monster flood of Pakistan in 2022 caused losses worth USD 30 billion (Chughtai, 2022). With the ongoing climate change crisis and high population density, the risk of flash floods

is anticipated to increase in the future; thus, emphasizing the need for flash flood loss modelling to derive quantitative estimates of expected losses in monetary terms.

While progress has been made related to fluvial flood loss models for households (Lüdtke et al., 2019; Thieken et al., 2008), there remains a limited understanding of the variables and mechanisms influencing flash flood losses. Unlike fluvial floods that have longer lead time and slower rise in water levels (Laudan et al., 2020), flash floods are characterized by rapid and

45 unprecedented rise in water levels. Moreover, due to high flow velocities, sediment transport, and shorter lead times, flash floods often cause more losses than fluvial floods. The sudden nature of flash floods makes them extremely difficult to predict (Dougherty and Rasmussen, 2020), necessitating loss modelling tailored to these events. Unlike floods caused by slowly rising water levels, and dyke breaches, flash floods exhibit heterogeneity in hazard characteristics such as water depth, flow velocity, inundation duration, and contamination indicators (Kreibich and Dimitrova, 2010). Furthermore, earlier studies suggest

significant differences in the variables and processes that influence losses in different flood types (Mohor et al., 2020). Thus, understanding the flash flood process is crucial, despite our previous understanding of the losses caused by fluvial floods. A comprehensive understanding of the complex processes behind flash floods is essential to develop sustainable and cost-efficient flash flood risk management strategies.

Traditionally, flood loss estimation relied on univariate stage-damage functions (SDF) (Middelmann-Fernandes, 2010). To

55 improve the description of complex damage processes, the Flood Loss Estimation MOdel (FLEMO$_{ps}$) for the private sector, was developed as rule-based, multivariate, deterministic model (Thieken et al., 2008). Merz et al. (2013) and Sieg et al. (2017) introduced decision tree-based damage models that explicitly quantify uncertainty associated with both data variability and model structure uncertainty through bootstrap aggregation. Subsequently, Bayesian Networks were used (BN-FLEMO), enabling the modelling of complex flood loss processes through conditional probability relationships (Lüdtke et al., 2019;

Schoppa et al., 2020; Schröter et al., 2014; Vogel et al., 2018).

In parallel, various machine learning approaches have also been developed for flood loss estimation, including neural networks (Salas et al., 2023), random forests (Ghaedi et al., 2022), Bayesian regression (Mohor et al., 2021). Among these, Bayesian networks are particularly advantageous due to their probabilistic representation of conditional dependencies among multiple

variables, handle missing data, and model transferability (Schröter et al., 2014). Bayesian models enhance the understanding of flood loss dynamics by quantifying uncertainty and offering probabilistic estimates. For instance, Wagenaar et al. (2018) developed a regional and temporal transferable BN-FLEMO for microscale residential applications, which was later upscaled to mesoscale by Lüdtke et al. (2019). In addition to the FLEMO typology, various synthetic, multivariate, rule-based flood loss models have been proposed for fluvial flood contexts (Amadio et al., 2019; Dottori et al., 2016; Nofal et al., 2020; Sairam et al., 2020).

However, all these loss models were developed to simulate damage processes during fluvial floods. In this study, we present the first probabilistic flash flood loss model – Flood Loss Estimation Model affected by flash floods (FLEMO$_{flash}$) using a BN-based approach and gain new knowledge about flash flood damage processes based on the conditional probabilities among multiple influencing variables. The study identifies the important variables and underlying processes that govern the flash flood losses. Additionally, we examine the predictive performance of FLEMO$_{flash}$ model and compare it with conventional SDF models. Finally, we illustrate the effect of preparedness in controlling the extent of loss reduction.

## 2 Data and Methods

### 2.1 Multievent empirical data in different regions

FLEMO$_{flash}$ is built based on self-reported flash flood losses along with associated information of the affected companies and households. The data was collected through different surveys using computer-aided telephone interviews with representatives of affected companies and households following three highly damaging flash flood occurrences in Germany (Kellermann et al., 2020; Kreibich et al., 2017). These include the 2002 event in the Elbe catchment in Eastern Germany, the 2016 heavy precipitation event, and the most recent July 2021 event in Western Germany. The variables potentially influencing losses are extracted and homogenized from the datasets of the 2002, 2016, and 2021 flood surveys. The variables are grouped into five categories, as presented in Table 1 for companies and Table 2 for households, respectively. An overview of all variables, along with the corresponding survey questions, response categories, and details on variable construction, is presented in Table S1 and Table S2. The loss variable represented as relative loss ($rloss$), is defined as the ratio between the reported loss and replacement cost on the [0,1] interval. A value of 0 indicates no loss, while 1 indicates complete loss (Kreibich and Dimitrova, 2010; Schoppa et al., 2020; Sieg et al., 2017). For companies, losses are estimated for three categories of assets - buildings, equipment, and goods & stock. For households, losses are estimated for buildings and contents. Survey responses with no data on loss were excluded from the dataset.

### 2.2 Terrain analysis for the identification of flash flood cases

Since the surveys covered regions affected by both flash floods and riverine floods, we applied a terrain-based filtering to exclude riverine cases. We calculated the median slope within a 10 km radius around the location of each observation using

the Digital Elevation Model with a 90 meters resolution from the Shuttle Radar Topographic Mission. This process was repeated for 14 reference municipalities known to have experienced flash flood events in the past or described as prone to flash floods (see Table A1). The minimum value among the median slope from these 14 municipalities was then used as the slope threshold (see Table S4). Observations with slope values equal to or above this threshold were classified as flash flood cases. Other metrics, such as river basin concentration time, may indeed provide a more process-based characterization of flash flood potential. Nevertheless, we used slope alone as a pragmatic solution that balances two competing needs: maintaining physical relevance in identifying flash flood prone companies and households, and retaining a sufficient number of data points for robust model development.

**Table 1. List of variables for companies. The variable type stands $C$ for continuous, $O$ for ordinal, and $N$ for nominal. An overview of all variables, along with the corresponding survey questions, response categories, and details on variable construction, is presented in Table S1.**

| | Predictors | | Type and range | |
|---|---|---|---|---|
| **Hazard** | $wd$ | Water depth | $C$: | 0 cm to 963 cm above ground |
| | $d$ | Inundation duration | $C$: | 0 to 1440 h |
| | $v$ | Velocity indicator | $O$: | 1 = low flow to 3 = torrential flow |
| | $con$ | Contamination | $O$: | 0 = no contamination to 4 = heavy contamination |
| **Preparedness/ Emergency response** | $wt$ | Warning lead time | $C$: | 0 to 240 h |
| | $ws$ | Early warning source | $O$: | 0 = no warning to 4 = official warning through authorities |
| | $ew$ | Early warning received | $N$: | 0 = no, 1 = yes |
| | $me$ | Emergency measures undertaken | $N$: | 0 = no, 1 = yes |
| | $ep$ | Emergency plan | $N$: | 0 = no, 1 = yes |
| | $kh$ | Knowledge about hazard | $N$: | 0 = no, 1 = yes |
| | $ms$ | Emergency measures success | $O$: | 0 = no measure undertaken, 1 = completely ineffective to 3 = very effective |
| **Precaution** | $fe$ | Flood experience | $O$: | 0 = no experience to 5 = recent flood experience |
| | $pr$ | Precaution indicator | $O$: | 0 = no precaution, 1 = medium precaution, 2 = very good precaution. |
| | $in$ | Insurance | $N$: | 0 = no, 1 = yes |
| **Company characteristics** | $sec$ | Sector | $O$: | 1 = Agriculture, 2 = Manufacturing, 3 = Trade, 4 = Finance, 5 = Services |
| | $ss$ | Spatial situation | $O$: | 1 = several buildings, 2 = entire building, 3 = one or more floors, 4 = less than one floor |
| | $own$ | ownership | $O$: | 1 = building owned, 2 = rented, 3= partly owned/ partly rented |
| | $emp$ | Number of employees | $C$: | 1 to 920 |
| | $sp$ | Size premise | $C$: | 10 to 69000 m$^2$ |

2021 data source: Survey "Flooding in Germany in July 2021: Damage of companies", German Research Centre for Geosciences, Deutsche Rückversicherung AG, 2022

2016 data source: Survey "Pluvial Flooding and Flash Floods in May/June 2016: Damage of companies", University of Potsdam, German Research Centre for Geosciences, Deutsche Rückversicherung AG, 2017.

2002 data source: Survey "Flooding in Germany in August 2002: Damage of companies", German Research Centre for Geosciences, Deutsche Rückversicherung AG, 2003

**Table 2. List of variables for households. The variable type stands $C$ for continuous, $O$ for ordinal, and $N$ for nominal. An overview of all variables, along with the corresponding survey questions, response categories, and details on variable construction, is presented in Table S2.**

| | Predictors | | Type and range |
|---|---|---|---|
| **Hazard** | $wd$ | Water depth | $C$: 245 cm below ground to 700 cm above ground |
| | $d$ | Inundation duration | $C$: 1 to 1440 h |
| | $v$ | Velocity scaled | $O$: 0 = no flow to 6 = torrential flow |
| | $hs$ | Human stability | $O$: 1 = person can stand effortlessly in calm water to 3 person would have been swept away; 4 = too deep to stand |
| | $con$ | Contamination | $O$: 0 = no contamination to 4 = heavy contamination |
| **Preparedness/ Emergency response** | $ew$ | Early warning received | $N$: 0 = no, 1 = yes |
| | $wt$ | Warning lead time | $C$: 0 to 168 h |
| | $ws$ | Warning source | $O$: 0 = no warning to 4 = official warning through authorities |
| | $ke$ | Knowledge about emergency action | $O$: 1= It was completely unclear to me to 6= It was completely clear to me |
| | $me$ | Emergency measures undertaken | $N$: 0 = no, 1 = yes |
| | $mu$ | Number of emergency measures undertaken | $O$: 0 = no measures undertaken to 13 = all measures undertaken |
| **precaution** | $fe$ | Flood experience | $O$: 0 = no experience to 5 = recent flood experience |
| | $pw$ | Precaution indicator | $O$: 0 = no/low precaution, 1 = medium precaution, 2 = very good precaution. |
| **building characteristics** | $fa$ | Building footprint area | $C$: 5 to 1000 m$^2$ |
| | $b$ | Basement | $N$: 0 = No basement, 1 = Partial basement, 2 = Full basement |
| **socio-economic status** | $per$ | Household size, i.e. number of persons | $C$: 1 to 12 people |
| | $chi$ | Number of children (< 14 yr) | $C$: 0 to 9 |
| | $eld$ | Number of elders (> 65 yr) | $C$: 0 to 4 |
| | $inc$ | Monthly net income in classes | $O$: 1 = below 500 EUR to 6 = 3000 EUR and more |
| | $socp$ | Socioeconomic status according to Plapp, (2003) | $O$: 3 = very low socioeconomic status to 13 = very high socioeconomic status |

2021 data source: Survey "Flooding in Germany in July 2021: Damage of private households", University of Potsdam, data collection within the KAHR-project, funded by the German Ministry of Education and Research (BMBF, contract 01LR2102I), approved by the ethical committee of the University of Potsdam (60/2022)

2016 data source: Survey "Pluvial Flooding and Flash Floods in May/June 2016: Damage of private households", University of Potsdam, German Research Centre for Geosciences, Deutsche Rückversicherung AG, 2017.

2002 data source: Survey "Flooding in Germany in August 2002: Damage of private households", German Research Centre for Geosciences, Deutsche Rückversicherung AG, 2003.

The final database available for developing loss models consists of 241, 379, 355, 1131, 1448 observations for company building, equipment, goods & stock, household building, and contents, respectively. The percentage of missing data for different variables in each of the asset is summarized in Fig S1 (companies) and Fig S2 (households). To maximise the amount of training data for model building, we employed the $k$ nearest neighbour technique to impute the missing data. We tested a range of $k$-neighbours for our datasets ($k$ =1,3,5,7,9) and selected the value with best performance.

## 2.3 Machine Learning-based feature selection

Flood damage processes vary by region, flood type, and asset type (Mohor et al., 2020; Sairam et al., 2021; Wagenaar et al., 2018). To derive the drivers of flash flood losses, this study adopts a data-driven feature selection approach to the empirical data. Feature selection involves identifying variables that have the highest influence on the target variable (i.e. relative loss). We train multiple models – nonlinear models: Random Forest (RF), Extreme Gradient Boosting (XGBoost), and linear model: Elastic Net (EN).

RF is an ensemble machine learning method primarily used for classification and regression tasks, developed by Breiman (2001). RF generates an ensemble of decision trees, each trained on a random subset of the data using bootstrap sampling. At each node within these trees, a random subset of features is considered for splitting. The final prediction for a given input is obtained by averaging the predictions from all individual trees. This approach helps RF reduce overfitting and enhances the model's generalization ability. XGBoost, similarly to RF, is an ensemble learning algorithm that benefits from a decision tree-based structure. However, the key difference compared to RF is that in XGBoost, each tree corrects the errors from the previous ones. The process starts with a simple model and iteratively adds trees that focus on the residuals or errors made by the existing ensemble. With its efficient implementation, XGBoost demonstrates superior performance and handles large-scale data more effectively than RF (Chen and Guestrin, 2016). While RF and XGBoost are non-linear models, EN is a regularization technique used in linear regression, combining both Lasso (L1) and Ridge (L2) regularization penalties. It effectively addresses multicollinearity in datasets by shrinking the less influential predictors toward zero (Lasso) while additionally providing some degree of regularization to prevent overfitting (Ridge). EN's ability to handle correlated features and select relevant predictors makes it a valuable tool in regression tasks (Zou and Hastie, 2005).

During training, we employed a nested cross-validation framework with 10 splits and 10 repeats, resulting in a total of 100 evaluations. We selected the best set of hyperparameters, which obtained the least mean absolute error, which was then applied to the final feature selection. From each resulting final model, we derived the feature importance. Next, we calculated each variable's weighted feature importance and overall rank. The final selection of the variables (Fig 1) is elaborated upon in the results section.

## 2.4 Probabilistic FLEMO_{flash} development

Based on the identified features, a multivariate probabilistic Flood Loss Estimation MOdel (Bayesian Network – BN) is calibrated for predicting flash flood losses (Jensen and Nielsen, 2007; Kitson et al., 2023; Scutari and Denis, 2021). BNs are probabilistic graphical models where the predictor and target variables are connected through a directed acyclic graph (DAG). Each variable is depicted as a node, and related nodes are connected through arcs. These connections allow for the estimation of conditional probability distributions, facilitating an understanding of the underlying processes and probabilistic estimation of rloss (Vogel et al., 2018). The continuous variables are discretized using an equal frequency discretization approach (Scutari and Denis, 2021), and a discrete BN is formulated. The number of bins is determined based on the model performance against

different numbers of bins – the number of bins resulting in the best model performance is chosen. The specification of a discrete BN involves defining a set of variables $(X_1, \ldots X_n)$, constructing a DAG representing the probabilistic dependencies among variables, and obtaining the conditional probability distribution $P\left(\frac{X_i}{parents(X_i)}\right)$ for each variable $(X_i)$ in the DAG, where $parents(X_i)$ denotes the parents of node $X_i$ in the DAG. The final joint probability distribution for the set of variables connected in a discrete BN is formulated as (Pearl, 1988):

$$P(X_1, \ldots, X_n) = \prod_{i=1}^{n} P\left(\frac{X_i}{parents(X_i)}\right) \tag{1}$$

Within the predicted bins of the discrete BN ($rloss$ bins), we fit a continuous distribution by applying weighted sampling to the empirical loss data, resulting in a smoothed representation of the loss distribution (Schoppa et al., 2020). For further details on the BN structure learning we refer to Text S1 and Figure A1.

### 2.4.1 Comparison to stage damage function

We compared FLEMO$_{flash}$ to a univariate stage-damage function (SDF), a conventional model in flood loss estimation (Gerl et al., 2016). We implemented the linear functional form of deterministic SDF (SDF-D) and probabilistic SDF (SDF-P) to assess the added value of the multivariate and probabilistic model. SDF-D is formulated as:

$$rloss_i = \alpha + \beta(wd)_i + \varepsilon_i \tag{2}$$

where $rloss_i$ is the relative loss for a given water depth $(wd_i)$. $\alpha$, $\beta$ and $\varepsilon_i$ are the intercept, regression coefficient, and error of observation $i$, respectively. Further, to implement the SDF-P, we assume that the relative loss follows a zero-and-one-inflated Beta distribution (Schoppa et al., 2020):

$$Y_i \ BEINF(\lambda, \gamma, \mu_i, \phi) \tag{3}$$

$$logit(\mu_i) = \alpha + \beta(wd)_i \tag{4}$$

In the above equation 3, we only predict the μ, whereas other distribution parameters are assumed constant across the observations. SI (Table S3) contains further information on the prior choice for model parameters as well as specification for Markov chain Monte Carlo sampling.

### 2.4.2 Model validation and sensitivity

We evaluate the performance of the FLEMO$_{flash}$, SDF-D, and SDF-P models individually for the different types of assets in both, companies and households. We employed 10-fold cross-validation, which was repeated (n=100) with independent random seeds to obtain robust estimates. To address parameter sensitivity of FLEMO$_{flash}$ model performances due to Bayesian network structure learning, we systematically evaluated three critical factors:

1.  Number of predictors (f1-f5): Section 2.3 identifies the ensemble-based important predictors, and the top five were used to develop the BN model. We demonstrated the model performance with varying number of predictors.

2.  Number of discretization bins (b3-b8): Continuous variables were binned using quantile-based stratification.

3.  Number of neighbours (k1-k9): Missing data were imputed with k-nearest neighbours.

For each combination, we validated the model for each cross-validation fold using three performance metrics: mean absolute error (MAE), continuous ranked probability score (CRPS), and mean bias error (MBE) (Gneiting and Katzfuss, 2014; Jensen and Nielsen, 2007; Krüger et al., 2021; Schoppa et al., 2020). Detailed information on the validation procedure and three scores used to compare the models are provided in SI.

**3 Results and Discussion**

**3.1 Drivers of flash flood losses**

An ensemble of linear and non-linear machine learning models ensures that both linear and non-linear relationships between the predictors and flood loss are captured. Water depth emerges as the most important predictor of damage across all asset types (Fig 1), which is consistent with previous loss models (Kreibich et al., 2010; Merz et al., 2013; Schoppa et al., 2020;
Sieg et al., 2017; Thieken et al., 2008). For companies, emergency measures success and number of employees are also significant factors influencing the flood loss estimation. Among other flood characteristics, duration is ranked fourth for building (Fig. 1a), velocity is ranked third for equipment (Fig. 1b), and contamination is the fifth most significant driver for goods & stock (Fig. 1c), respectively. In case of households, human stability and contamination are most important hazard variables after water depth.

The significance of water depth and emergency measures has also been emphasized by Hasanzadeh Nafari et al. (2016) in case of fluvial flood losses. Exposure variables such as number of employees significantly influence company losses. Additionally in case of households, losses are more influenced by variables representing flood vulnerability such as knowledge about emergency action (Fig. 1d-e). These findings are in line with the previous studies (Kreibich et al., 2005; Sairam et al., 2019; Zander et al., 2023) that highlight the potential for adaptation measures to reduce flood losses. By identifying the varied drivers
of flash flood losses, these results also emphasize the importance of multivariable loss estimation models that capture the interplay across these drivers and their influence on losses.

Although flow velocity has been identified as a significant contributor to flash flood losses (Kreibich and Dimitrova, 2010), it does not appear among the most significant factors in the current study. In our analysis, we represent hydrodynamic forces using velocity ($v$) and human stability ($hs$). While velocity provides a subjective yet direct measure of local strength of flow
current, human stability reflects on the perceived difficulty of standing in flood waters. As shown in Figure 1(d-e), human stability emerges as the second most influential factor affecting loss in households, indicating that the combined effect of water depth and velocity play a crucial role for the flash flood model.

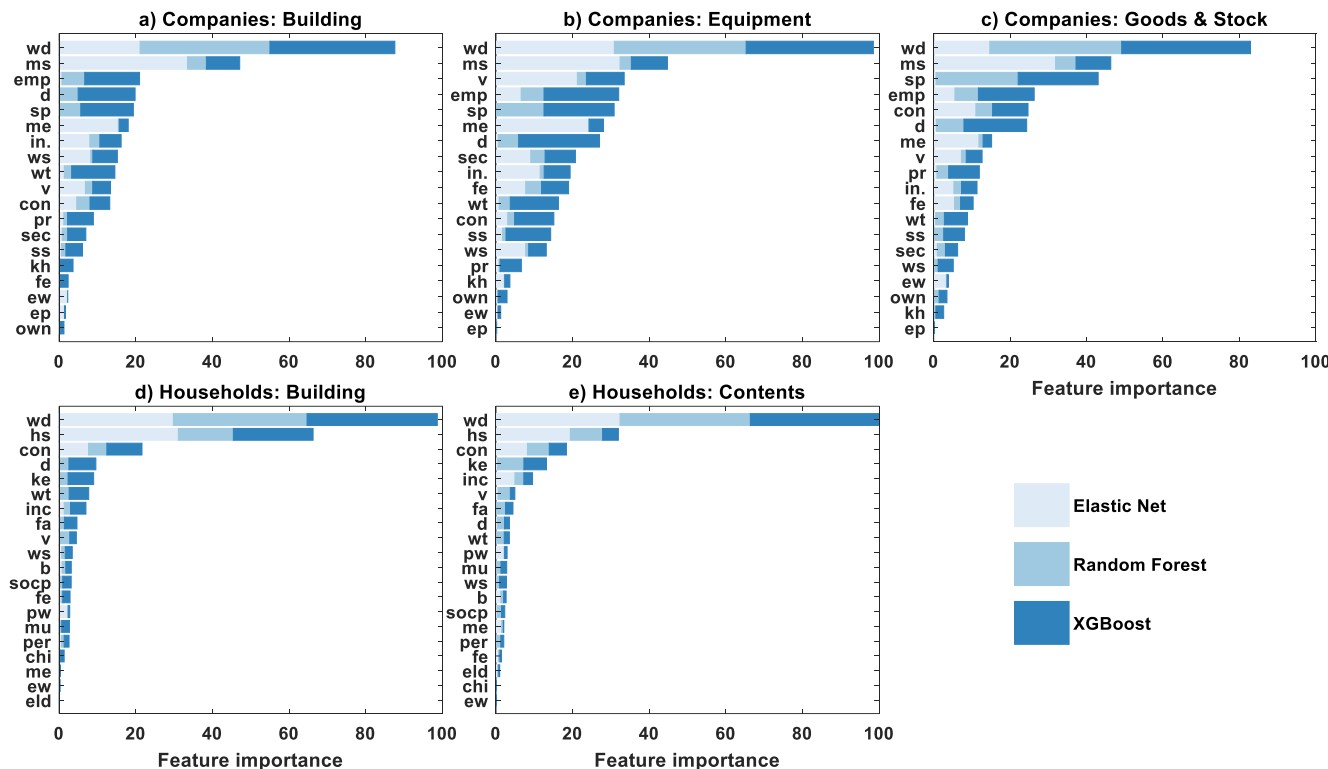

Figure 1. Illustration of the feature importance in predicting the relative loss of companies categorized into (a) Building, (b) Equipment, (c) Goods and Stock. Similarly, households categorized into (d) Building and (e) Contents. X-axis denotes the weighted importance derived from an ensemble approach, combining two non-linear models (Random Forest and Extreme Gradient Boosting) and one linear model (Elastic Net).

## 3.2 Probabilistic multivariate flash flood loss model

### 3.2.1 FLEMO_flash Bayesian network structure

The FLEMO_flash models are developed using a score-based structure learning algorithm and BN models are developed to capture the multivariate dependencies among variables. Data-driven BNs with the best performance were evaluated by domain experts to ensure consistency with the existing understanding about the underlying dynamics of flood loss processes (Fig. 2). The direction of the arrow represents an association between two variables but does not necessarily represent causality (Lüdtke et al., 2019; Sairam et al., 2020). Water depth emerged as the most important predictor for loss estimation across all asset types and is directly connected to the $rloss$ node in all BN structures (Fig. 2). In case of companies and households, the losses are influenced predominantly by water depth. Company characteristics (number of employees and size premise) significantly impact the losses for company assets (Fig. 2a-c). The measure success (ms) predictor is directly connected to $rloss$ in case of building and equipment (Fig 2a-b) and indirectly connected to $rloss$ through water depth for goods & stock (Fig. 2c). In households, building $rloss$ is directly connected to water depth, contamination, and knowledge about emergency action (Fig 2d). For contents, $rloss$ is directly connected to water depth and knowledge about emergency action, with contamination,

human stability, and income also playing important roles (Fig. 2e). Additionally, human stability, which is a factor of both depth and velocity, influences household losses through water depth. The direct connection of $rloss$ with measures success for companies' assets (Fig. 2a-c), and with knowledge about emergency action for households (Fig. 2d-e) highlights the significance in mitigating flood losses. We derived joint probability distribution for all asset types and the findings reveal trade-offs between preparation strategies for different target losses.

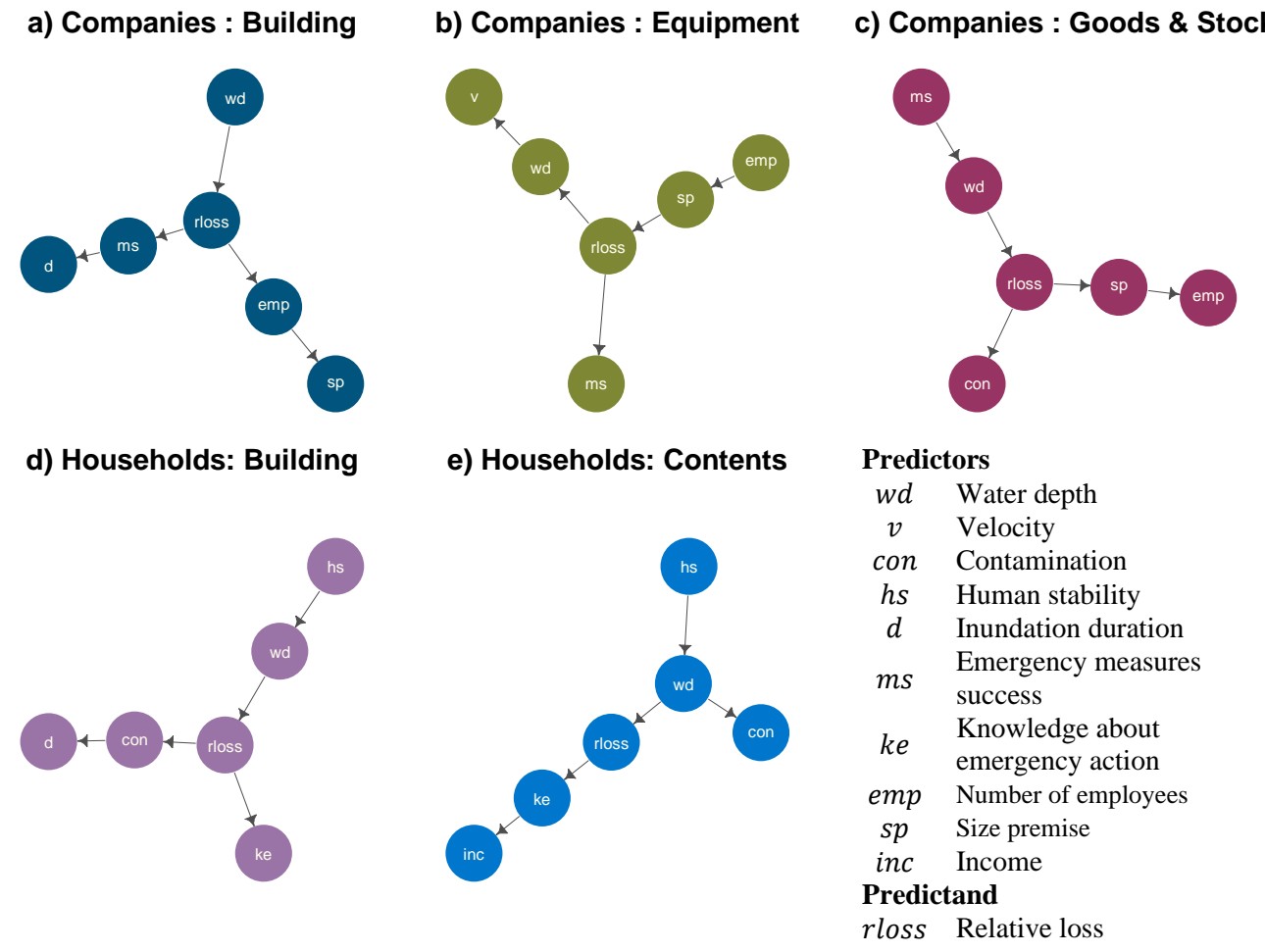

**Figure 2. Bayesian Network structures for (a) Companies: Building, (b) Companies: Equipment, (c) Companies: Goods and Stock, (d) Households: Building, (d) Households: Contents, obtained from score-based structure learning algorithm.**

### 3.2.2 Performance and comparison

The performance of the FLEMO$_{flash}$ BN structure was tested considering varying numbers of predictors, bins, and k-nearest neighbors. In the BN structure, null value was assigned to one predictor at a time (from lowest-5 rank to highest-1 rank based on feature importance – Fig 1) to examine the performance using the remaining variables (Fig 3). The performance metrics

(MAE, CRPS) vary with the number of predictors. The predictive performance improved with the number of predictors. For instance, the MAE values for all asset types decreased as the number of predictors increased. Based on the performance metrics, the optimum number of predictors was found to be five. However, the Markov Blanket of loss consists of two predictors (water depth and knowledge of emergency measures) for households: contents loss, and three predictors otherwise. Similarly, CRPS values also showed a declining pattern with increasing number of predictors, indicating better probabilistic predictions. The

MBE was relatively stable, suggesting that bias in the prediction did not significantly change with the number of predictors. Examining the performance with optimal predictors while modifying the number of bins, revealed significant differences for companies but not for households, which could be attributed to the fact that the number of data points for companies is relatively limited and more heterogenous (Schoppa et al., 2020) compared to households. Company buildings model (C:BUI) with too few bins tend to lose information, resulting in higher MAE and CRPS values. Conversely, the goods & stocks model

(C:GNS) performed better with fewer bins. For household models, MAE was not sensitive to the choice of bins, but model performance was slightly better with 8 bins. The MBE for company models showed fluctuations around zero, indicating some sensitivity to the choice of bins and potential bias.

The number of neighbours used for data imputation does not show any significant difference in predictive performance. The MBE showed minor variations, remaining close to zero across different numbers of neighbours, suggesting that the bias in the

250 predictions was not significantly affected by the imputation. The k-nearest neighbours (kNN) method of imputation assumes that the missing values can be inferred based on similarity of feature space. This may not hold equally well across variables, particularly for those with weak correlation to other features. To evaluate the robustness of the imputation process, we compared the distributions of variables before and after imputation and found them to be largely consistent (not shown for brevity). Nevertheless, the imputation process may still introduce uncertainty or reduce natural variability in the data. Future

studies could benefit from sensitivity testing using alternative imputation techniques and explore models that explicitly incorporate imputation uncertainty.

Overall, building assets in both company and household sectors performed better compared to other asset types. This superior performance is due to the effective inference of the relationship between water depth and building $rloss$ (Gerl et al., 2016; Lüdtke et al., 2019; Merz et al., 2013; Vogel et al., 2018). Additionally, the emergency response of mitigation measures for

companies and knowledge about emergency action for households also boosted performance.

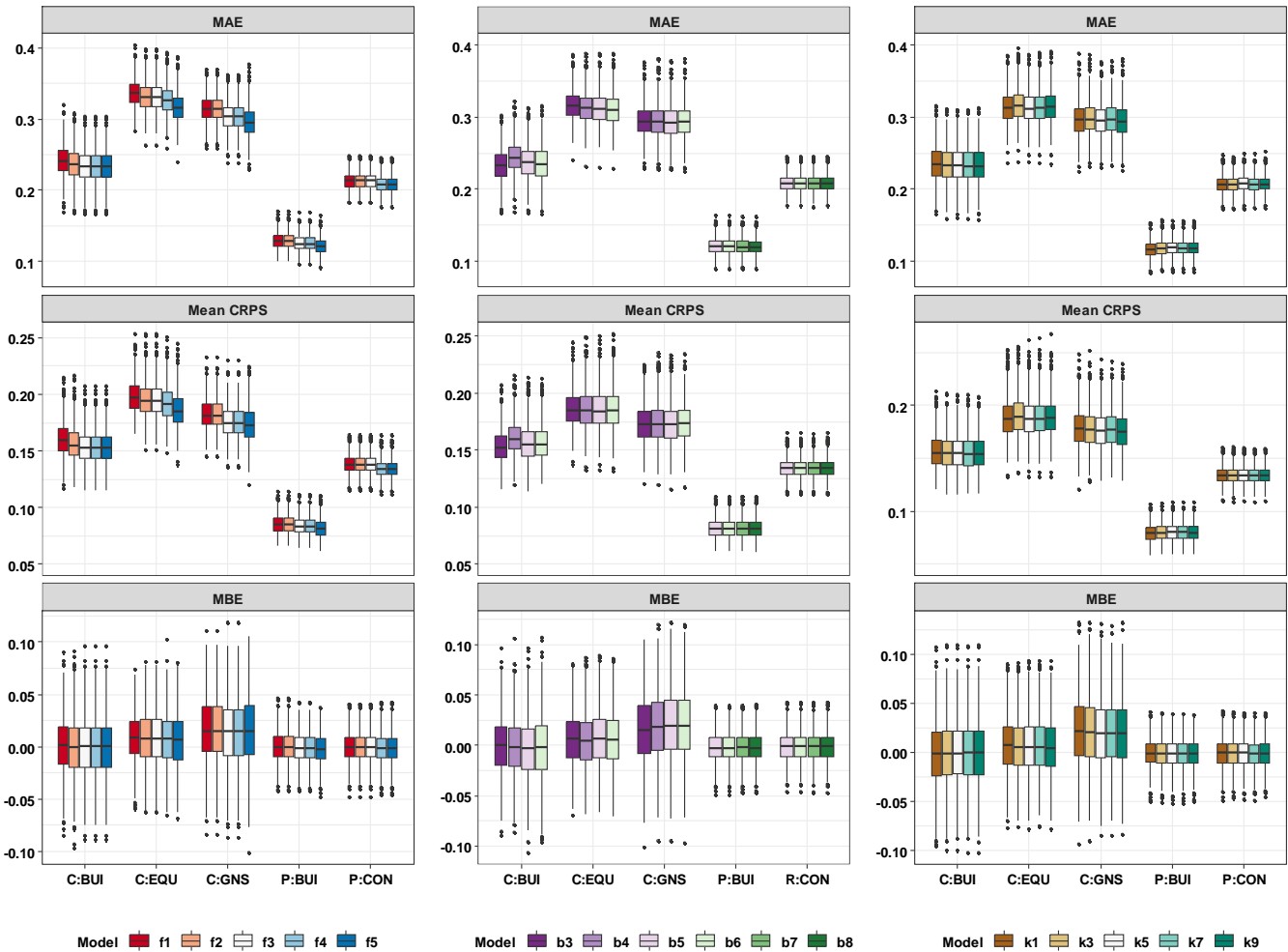

**Figure 3. Model sensitivity of FLEMO*flash* to number of predictors (f1-f5), bins (b3-b8), number of neighbours used for data imputation (k1-k9) evaluated using mean average error (MAE), continuous ranked probability score (CRPS), and mean bias error (MBE) for the five asset types (x-axis). Each bocplot summarizes 100 repetitions of a fivefold cross-validation (companies) and tenfold cross-validation (households) with randomized data partitioning. Best-performing configurations were identified through a sequential tuning process: first selecting the number of predictors based on the first panel, then optimizing bin count in the second panel with predictors fixed, and finally selecting the number of neighbours in the third panel with both previous parameters fixed. Best-performing configurations are: Companies – Buildings (C:BUI) f5, b6, k7; Companies – Equipment (C:EQU) f5, b6, k5; Companies – Goods and Stock (C:GNS) f5, b6, k9; Private Households – Buildings (P:BUI) f5, b8, k1; Private Households – Contents (P:CON) f5, b8, k3.**

The FLEMO*flash* model with the best performance, identified in Fig 3 was compared to SDF models (Fig 4). The FLEMO*flash* model consistently outperformed both the SDF-probabilistic and SDF-deterministic models across all five asset types due to its comprehensive representation of loss processes (Schröter et al., 2014). Study by Schoppa et al. (2020) also noted a similar observation reporting that probabilistic multivariate models performed better than univariate models for fluvial flood loss estimation. These findings highlight that FLEMO*flash* outperformed in estimating losses while also indicating the need for improvement to address biases in the predictions.

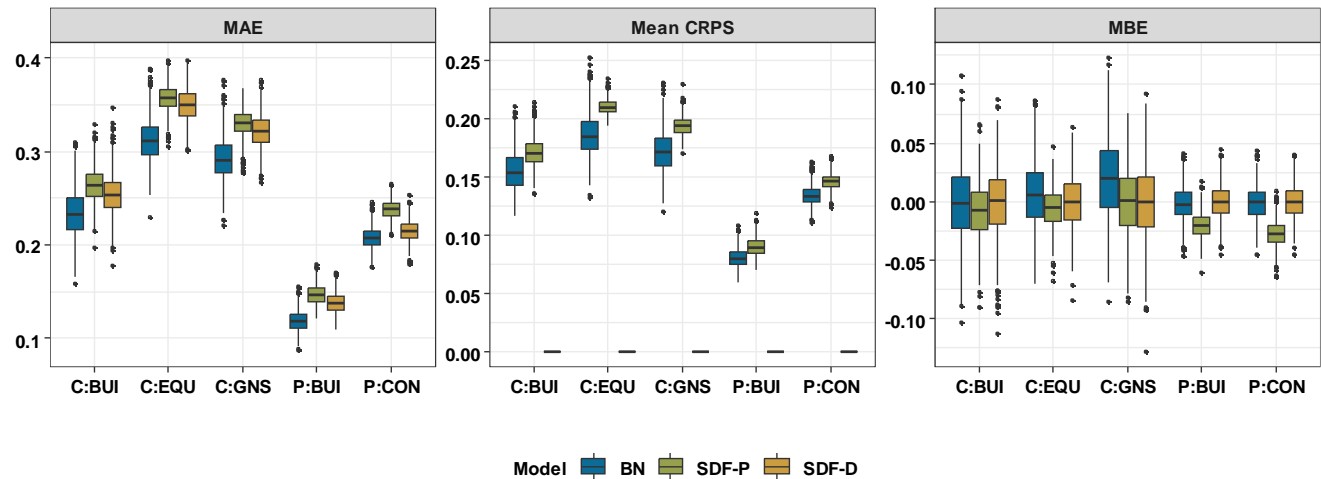

**Figure** 4. **Comparison of FLEMO**flash **predictive performance against SDF-probabilistic and SDF-deterministic using MAE, CRPS, and MBE for the five assets (x-axis). Each boxplot summarizes 100 repetitions of a fivefold cross-validation (companies) and tenfold cross-validation (households) with randomized data partitioning.**

### 3.3 Description of loss processes by FLEMOflash

The loss processes described by FLEMOflash is illustrated using the predictive density of predicted losses under scenarios of hazard, exposure and vulnerability. For brevity, this section primarily focusses on the FLEMOflash model for household buildings (Fig 5), with a similar interpretation extended to other asset types (Fig S3-S6). The nodes of the model comprise of water depth, human stability, inundation duration, contamination, knowledge about emergency action, and relative losses, each with 7, 4, 7, 5, 6, and 8 classes, respectively. The Conditional Probability Table (CPT) was populated with joint probabilities to find the predictive density of loss given the condition of other nodes. The conditional probability of *rloss* based only on water depth indicates a monotonic relationship. Shallow inundations are associated with very low losses, while deeper water substantially increases the probability of severe losses (Fig 5e). The highest probabilities are concentrated along the diagonal, confirming this trend. For instance, depths <0.28 m are most likely associated with very low losses (<0.017), whereas depths ≥2.3 m are strongly associated with high losses (> 0.42). Similar patterns of increasing loss probability with greater water depth are observed across all asset types (Fig. S3–S6). Water depth also influences human stability: while shallow flooding results in low instability, extreme depths markedly increase the probability of instability (0.54) (Fig. 5b, Fig. S6a).

Contamination emerges as another important driver of losses. In uncontaminated conditions (class 0), the probability of very low losses (<0.01) is high (0.82). Conversely, under severe contamination (class 4), the probability of very high losses (>0.427) increases to 0.30 (Fig. 5c), reflecting the destructive impact of oils, chemicals, and sewage entering buildings (Kreibich et al., 2005; Laudan et al., 2020). Households exposed to inundation lasting [13–50) hours showed a high probability of experiencing moderate contamination levels (classes 1–2). Knowledge about emergency action shows a strong mitigating effect. The CPT (Fig. 5d) demonstrates that households with low awareness (Ke ≤ 2) face a high probability of severe losses, whereas

households with very good knowledge (Ke ≥ 5) display a substantially higher probability of reduced losses. Comparable findings are observed for household contents (Fig. S6c). This agrees with Kreibich et al. (2021), who also reported that clear awareness of emergency actions substantially reduces damages. Importantly, socioeconomic status indirectly shapes vulnerability, as higher-income groups are more likely to report very clear knowledge of emergency actions after receiving warnings (Fig. S6b).

For companies (Figs. S3–S5), the CPT results reveal consistent patterns across buildings, equipment, and goods & stock. Smaller companies (with fewer employees or smaller premises) show higher probabilities of severe losses, whereas larger firms and premises are more strongly associated with lower loss outcomes (Figs. S3b, S4d, S5c). Across all asset types, the success of emergency measures emerges as a dominant factor, as unsuccessful measures are strongly associated with a high probability of severe losses (Figs. S3d, S4b, S5a). Contamination further amplifies losses, with severe categories linked to markedly higher probabilities of loss. Together, these results emphasize that hazard intensity (water depth, velocity, contamination), exposure (number of employees, size premises) and vulnerability factors (effectiveness of emergency measures) interactively determine relative losses for companies.

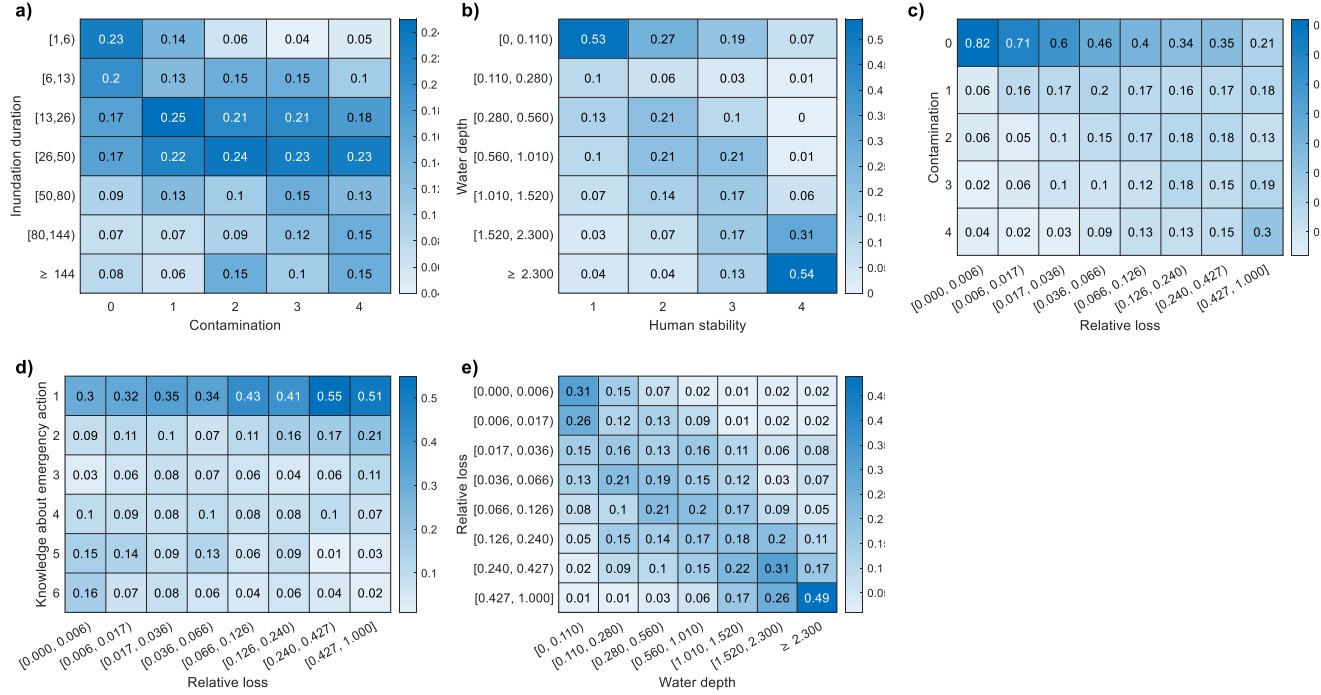

**Figure 5. Conditional probability table (CPT) of the Bayesian network for the residential building. Each heatmap illustrates the conditional probabilities of a child node given its parent node. Parent node states are shown along the x-axis, and child node states along the y-axis. Darker shades of blue indicate higher probability values. Numerical values are displayed in each cell, with an accompanying colorbar showing the probability scale. (a) Inundation duration given contamination (b) Water depth given human stability (c) Contamination given relative loss (d) Knowledge about emergency action given relative loss (e) Relative loss given water depth.**

### 3.3.1    Effect of preparedness

Through feature selection and Bayesian Networks we identified emergency measures success ($ms$) and knowledge about emergency action ($ke$) for companies and households respectively, as the significant variables (see Tables S2 and S3 for details on the questions and responses). Building on this, we conceptualised preparedness using these variables and categorized it into low, medium, and high levels. For companies, high preparedness was defined as having undertaken emergency measures that were perceived to be mostly or completely effective ($ms = 3$) and low preparedness ($ms = 1$) reflected low perceived effectiveness of such measures. For households, high preparedness was defined as having a clear understanding of emergency actions based on official warnings ($ke \geq 5$), and low preparedness reflected limited to no understanding of what to do ($ke \leq 2$). While preparedness has been extensively studied in the context of fluvial or riverine floods (Lüdtke et al., 2019; Schoppa et al., 2020; Wagenaar et al., 2018), its role in flash floods has not yet been systematically investigated. To address this gap, we applied the FLEMO$_{flash}$ model to derive predictive densities of $rloss$. Results were summarized using the median and associated uncertainty (25$^{th}$ and 75$^{th}$ percentiles) for selected combinations of hazard, exposure, and vulnerability conditions, rather than displaying the full predictive densities (Figure 6). For clarity of interpretation, Figure A1 illustrates step by step how predictive densities are derived from the prior and posterior distributions using kernel density estimation based on 1,000 resampled values, while Figure A2 provides an overview of the posterior and predictive densities across varying levels of measure success (preparedness) under same conditions of water depth and number of employees.

For company building, the incurred loss increases with increasing water depth. Considering the company characteristics, the relative loss experienced by companies with 9 to 12 employees is higher compared to those with 38 or more employees. When taking preparedness into consideration, we see its significance in reducing loss. For example, when considering only hazard & exposure, companies with 38 or more employees, inundated by water depths ranging from 1.85 m to 2.40 m above the ground, experience a $rloss$ of 0.38. However, with high preparedness (ms=3), this $rloss$ decreases to 0.20, representing a substantial 47% reduction. On the other hand, with low preparedness, the $rloss$ increases to 0.50, marking a 32% increase in estimated losses.

For household buildings, considering hazard and exposure with a water depth of $\geq 2.3$ m above ground and no contamination, the predicted $rloss$ is 0.22. However, with high preparedness ($Ke \geq 5$) this loss decreases to 0.05, reflecting a 77% reduction. Conversely, with low preparedness ($Ke \leq 2$), the $rloss$ rises to 0.27, a 23% increase. Preparedness also plays a crucial role in mitigating losses to household contents. For instance, when considering only water depths of  height 2.3 m above ground, the $rloss$ is 0.38. With high preparedness, this loss drops to 0.17, a 55% reduction. In contrast, with low preparedness, the $rloss$ increases to 0.44, representing a 16% rise. In both companies and households, losses are consistently higher with low preparedness and lower with high preparedness. The extent of this difference varies depending on hazard and exposure characteristics. These findings highlight the effect of preparedness in reducing the risk of flash flood risk.

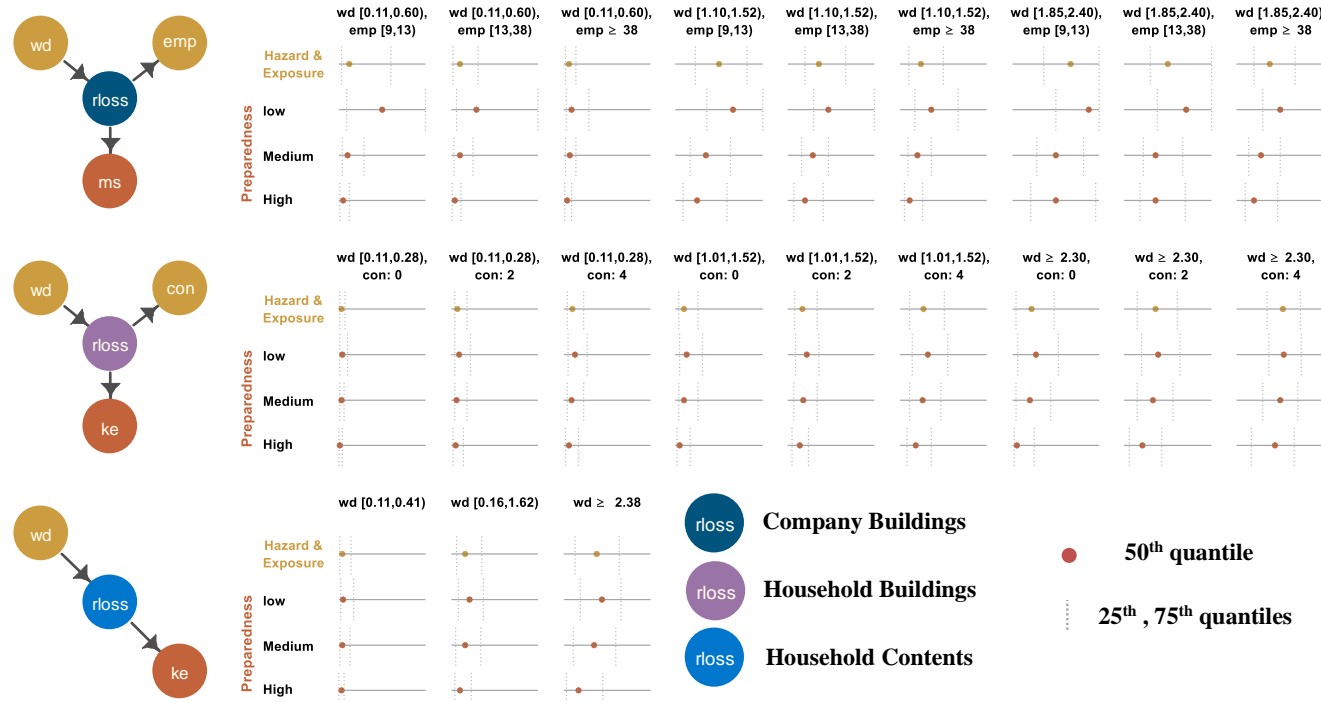

**Figure 6. FLEMO**flash **application for company buildings, private household buildings and contents, considering relative loss Markov blankets. The first row in each panel shows the probabilistic predictive density of relative loss on the interval [0,1] based on the specific scenarios of hazard and exposure combination. The second to fourth rows in each panel illustrate the changes in relative loss with different levels of preparedness for the given hazard and exposure combinations.**

Preparedness is crucial in mitigating potential losses (Barendrecht et al., 2020; Berghäuser et al., 2023; Bubeck et al., 2013; Sairam et al., 2019; Surminski and Thieken, 2017). Residents with high levels of preparedness are more likely to take effective emergency measures, thereby reducing the severity of flood loss (Kreibich et al., 2005; Sairam et al., 2019). Despite its importance, the way preparedness is conceptualized in this study has certain limitations. Specifically, the variable does not capture which exact actions respondents undertook. Therefore, it would be misleading to speculate particular actions directly resulted in reduced losses. While the specific actions likely varied across respondents, empirical evidence indicates that having clear knowledge of emergency action generally contributes to better preparedness, consistent with previous findings (Kreibich et al., 2021).

## 4 Conclusions

This study introduces FLEMO_flash - new set of multivariate flash flood loss models for estimating relative losses for companies and households. The machine learning feature selection identified key loss drivers comprising hazard, company characteristics, and emergency response. Results further demonstrate that preparedness can substantially reduce losses. In extreme hazard scenarios, losses were reduced by nearly half for larger companies, and households that received timely warnings and has clear knowledge were able to reduce building losses by 77% and contents losses by 55%. While FLEMO_flash already provides a

robust tool to support risk analyses, and impact-based forecasting, future developments could further strengthen its applicability by integrating complementary hydrological indicators (e.g., basin concentration time), incorporating building-level susceptibility factors (e.g., construction materials, structural condition, floor count), and expanding the empirical database by including high loss observations and more diverse geographic regions.

**Table A1. Overview of 14 municipalities affected by past flash flood events.**

| Name of Municipality | Latitude (N) | Longitude (E) | Reference (including research papers, official reports, municipal flash flood maps, media coverage of past events) |
|---|---|---|---|
| Triftern | 48.3957 | 13.0060 | (LfU, 2017), Thieken et al. (2022) |
| Simbach am Inn | 48.2869 | 13.0113 | Hübl et al., (2017), (LfU, 2017), Thieken et al. (2022) |
| Obernzenn | 49.4492 | 10.4886 | (LfU, 2017), Thieken et al. (2022) |
| Künzelsau | 49.2802 | 09.7378 | Mühr et al. (2016), Thieken et al. (2022) |
| Julbach | 48.2547 | 12.9313 | (LfU, 2017), Thieken et al. (2022) |
| Forchtenberg | 49.2799 | 09.5149 | Mühr et al. (2016), Thieken et al. (2022) |
| Flachslanden | 49.4081 | 10.5205 | (LfU, 2017), Thieken et al. (2022) |
| Braunsbach | 49.2007 | 09.7873 | (Bronstert et al., 2018), Thieken et al. (2022) |
| Ansbach | 49.2888 | 10.5553 | (LfU, 2017), Thieken et al. (2022) |
| Stadtallendorf | 50.8308 | 09.02447 | AVOSS Test Municipality (https://www.avoss.uni-freiburg.de/testgebiete). Past event (https://www.feuerwehr-wetter.de/informationen/buergerinformationen/starkregen.html) |
| Grafschaft | 50.5752 | 07.0852 | AVOSS Test Municipality (https://www.avoss.uni-freiburg.de/testgebiete). Past event( https://hochwasser-grafschaft.de/?p=936) |
| Herrstein | 49.7845 | 07.3461 | AVOSS Test Municipality (https://www.avoss.uni-freiburg.de/testgebiete). Past event (https://fachtagung-funke.de/wp-content/uploads/2024/06/6_Fuhr_Eisatzbericht-Herrstein_2018.pdf) |
| Otting | 48.8801 | 10.7978 | AVOSS Test Municipality (https://www.avoss.uni-freiburg.de/testgebiete). Past event (https://www1.wdr.de/nachrichten/westfalen-lippe/aufraeumarbeiten-starkregen-ottfingen-100.html) |
| Emmendingen | 48.1225 | 07.8623 | AVOSS Test Municipality (https://www.avoss.uni-freiburg.de/testgebiete). Municipal flash flood maps (https://www.emmendingen.de/leben-umwelt/vorsorge-krise/starkregen) |

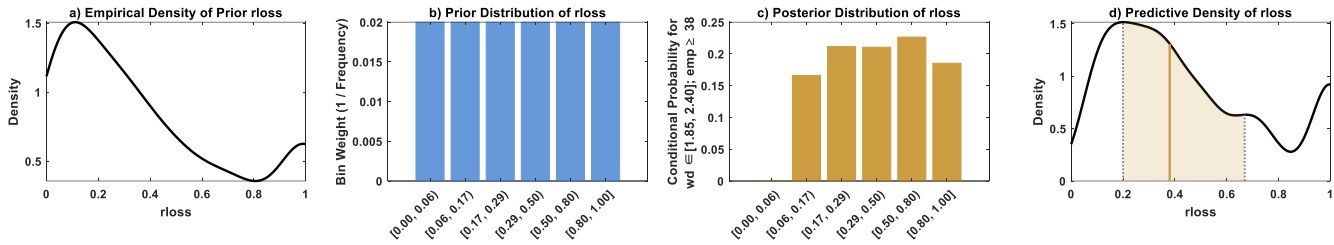

**Figure A7: Visualizations of the prior, posterior, and predictive distributions of rloss (a) Empirical kernel density estimate of the prior rloss based on collected data (b) Prior distribution of rloss represented as bin weights (inverse frequency) across discretized intervals (c) Posterior distribution of rloss conditioned on wd ∈ [1.85, 2.40) and emp ≥ 38 (d) Predictive distribution of rloss generated by resampling 1000 values using the prior bin weights and the posterior probabilities. The solid vertical line indicates the median (50th percentile), while the dotted vertical lines represent the 25th and 75th percentiles, representing the predictive uncertainty. Shaded area highlights the interquartile range.**

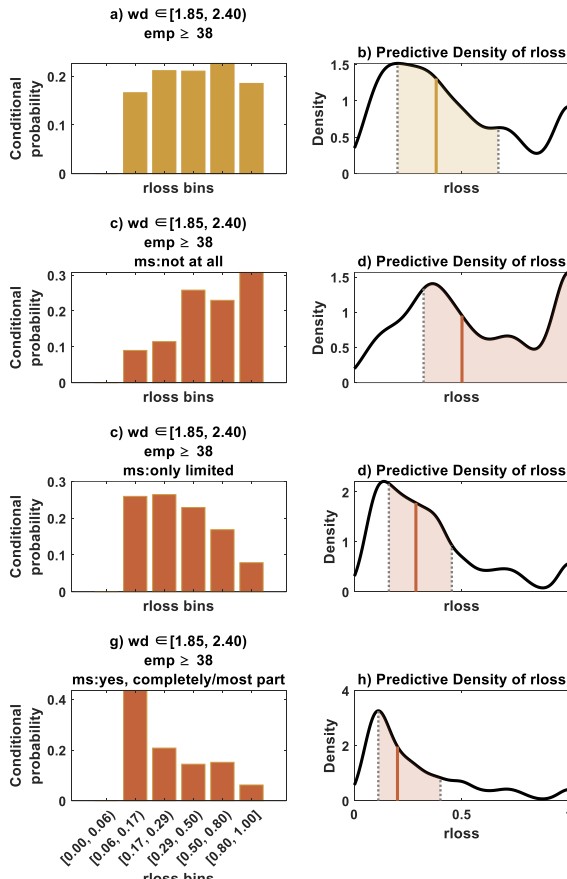

**Figure A8: (a) Posterior distribution and (b) predictive density of relative loss (rloss) under condition of water depth (wd) ∈ [1.85, 2.40) and number of employees (emp) ≥ 38. (c–h) Posterior and predictive distributions of rloss for varying levels of measure success (ms): Subplots c, e, g present posterior distributions of rloss under three ms conditions — not at all, only limited, and yes – completely/most part — with wd ∈ [1.85, 2.40) and emp ≥ 38. Subplots d, f, h shows corresponding predictive densities, estimated using kernel density estimation from resampled values (n = 1000). In each density plot, the solid vertical line marks the median (50th percentile), while dotted vertical lines indicate the 25th and 75th percentiles, with shaded regions representing the uncertainty. The sequence from top to bottom illustrates increasing levels of preparedness.**

## Acknowledgements

This research has been supported by the German Federal Ministry of Education and Research (BMBF) within the framework of the AVOSS project (grant no. FKZ 02WEE1629C). Apoorva Singh acknowledges Helmholtz Information & Data Science Academy (HIDA) for providing financial support for the research stay at the Section Hydrology, GFZ. Nivedita Sairam is funded by the BMBF project "HI-CliF", FKZ: 01LN2209A. Collection of the 2021 private household data was undertaken by the Geography and Disaster Risk Research Lab, University of Potsdam within the KAHR-project, funded by BMBF (contract 01LR2102I). Collection of the 2021 company data was undertaken by Section Hydrology, GFZ and Deutsche Rückversicherung AG, funded by the GFZ-HART initiative and Deutsche Rückversicherung AG. Collection of the 2016 data was undertaken within the DFG Research Training Group "NatRiskChange" (GRK 2043/1), funding is gratefully acknowledged, as are additional funds provided by GFZ and the Deutsche Rückversicherung AG. Collection of the 2002 data was undertaken within the German Research Network Natural Disasters (DFNK), in cooperation between GFZ and the Deutsche Rückversicherung AG, we thank the Deutsche Rückversicherung AG and BMBF (01SFR9969/5) for financial support. We would like to thank the anonymous reviewers for their constructive feedback on the article.

## Data Availability Statement

The survey data is partly accessible at the German flood damage database, HOWAS21 (http://dx.doi.org/10.1594/GFZ.SDDB.HOWAS21)

## Author contribution

**Conceptualization**: AS, RKG, KRS, NS, HK; **Methodology**: AS, RKG, NS, KRS, AB, MF, HK; **Data curation**: AS, RKG, MF; **Writing- original draft**: AS, RKG; **Writing- review & editing**: RKG, NS, KRS, HK; **Visualization**: AS, RKG; **Supervision**: CTD, HK

## Competing interests

The author Heidi Kreibich is a member of the editorial board of Natural Hazards and Earth System Sciences.

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
