# Peer review of "FLEMOflash - Flood Loss Estimation MOdels for companies and households affected by flash floods"

_EGUsphere, 2025_

## Referee Comment (RC2)

Journal: NHESS
Title: **FLEMOflash - Flood Loss Estimation MOdels for companies and households affected by flash floods**
Author(s): Singh et al.
MS No.: egusphere2025-1512
MS Type: Research Article
**Iteration: First review**

The paper introduces **FLEMOflash**, a novel multivariate probabilistic Flood Loss Estimation Model tailored for flash floods. The model builds on survey data collected after flash flood events in 2002, 2016, and 2021 in Germany, encompassing both affected companies and households. FLEMOflash employs a data-driven feature selection approach alongside Bayesian networks to derive probabilistic loss estimates. The topic clearly falls within the scope of the journal, and the manuscript is generally well written and well organised. However, I have concerns regarding some of the underlying assumptions of the model, which, in turn, raise doubts about its validity for reliably estimating flash flood damage. I believe the authors should provide a more robust justification for their hypotheses to strengthen the credibility and robustness of their results.

Below, I first present general concerns, followed by more specific comments.

**General concerns**

1)      My first concern relates to the criteria used for identifying flash flood events (and then data to be implemented to derive the model). Specifically, I find the use of average slope as a proxy problematic. While slope may influence local flow velocity, it does not adequately capture the main defining characteristic of flash floods — their rapid onset and short lead times. This concern is further supported by the reported warning lead times in Tables 1 and 2, which range from 0 to 240 hours and 0 to 168 hours, respectively. These values appear inconsistent with typical flash flood dynamics, where lead times are often just a few hours.

Additionally, the use of a low-resolution DEM may not provide the accuracy needed to derive reliable slope estimates at the point observation scale.

Have the authors considered using the *concentration time* of the river basin where the observations are located as a more physically meaningful proxy for flash flood potential? This could provide a better indication of response time and be more consistent with established hydrological understanding of flash flood processes.

2)      My second concern relates to the set of explanatory variables used in the model. One of the primary damage mechanisms in flash flood events is structural damage, which is strongly influenced by the physical vulnerability of affected buildings. However, the model does not appear to include variables that capture this aspect, such as construction material, number of floors, or level of maintenance — all of which significantly affect a building's susceptibility to structural damage.

While I understand that the set of variables was likely constrained by the information collected through the survey, I would like to know whether the authors considered integrating ancillary data to address these critical gaps. For example, building-level data from national censuses or geoportals could provide valuable proxies for physical vulnerability.

Inclusion of such information could improve the explanatory power and practical relevance of the model, particularly in contexts where decisions rely on nuanced understanding of asset-specific vulnerabilities.

3)      A third concern regards obtained results, especially in terms of damage mechanisms. I would have expected to observe a significant influence of flow velocity or, at least, of the hydrodynamic force associated with the flow but this is not the case.

4)      All the concerns mentioned above converge in the results obtained, particularly in the relative loss estimates provided by the model. These estimates range between 0.2 and 0.5, even for high water depths (around or above 2 meters). Such values are comparable to those typically produced by models for riverine

floods (see, e.g., FLEMO ps), which raises doubts about the model's ability to capture the distinctively more destructive nature of flash floods.

**Specific comments**

Table 1 and Table 2 → the meaning of some variables is not clear. For instance, does emergency plan refer to the existence of a municipal emergency plan or a company emergency plan? Which is the meaning of the precaution indicator? Which are the emergency measures considered? I suggest including an explanatory table in the supplementary material

Line 102-104
*"To maximise the amount of training data for model building, we employed the nearest neighbour technique to impute the missing data. We tested a range of $k$-neighbours for our datasets ($k$ =1,3,5,7,9) and selected the value with best performance"* → while this could be a good option for spatially correlated variables such as velocity and warning lead time (after verifying that the distance between points is limited), it may lead to misleading assumptions for other missing variables. For example, variables such as *in*, *sp*, and *sec* (for companies) or *ke*, *fa*, and *b* (for buildings) are not necessarily spatially correlated. It would be helpful if the authors could provide a more thorough discussion on this point, particularly addressing the potential limitations and implications of their imputation strategy for these types of variables.

Section 3.1
The meaning of two CPTs in the table (d-con, wd-hs) should also be discussed. Moreover, I think this section should be expanded discussing results for all damage components (i.e. companies BUI, EQU, GNS and household CON), even without reporting all the CPTs.

Figure 5 → I think that results explanation will be supported if each CPT is identified with a letter

Line 289- 296
*"The integration of knowledge about emergency action into the FLEMOflash model alongside water depth and contamination provides a comprehensive understanding of how preparedness can mitigate loss during flash floods. Knowledge about emergency action is categorized into six classes, ranging from 1 (low knowledge) to 6 (high knowledge). The CPT clearly illustrates that a high level of emergency action knowledge can significantly reduce loss (Fig 5e). Specifically, when households doesn't knew what to do (1), there is a high likelihood of incurring higher loss. Conversely, when households with good preparedness (> 4), the incurred loss significantly decreases. Residents with high levels of preparedness are more likely to take effective emergency measures, thereby reducing the severity of flood loss"* → Knowing what to do does not necessarily imply that individuals will take action. Do the authors have any insight into why this variable appears to be significant in the model, potentially even more so than the actual implementation of protective measures (me, mu)?

**Minor comments**

Line 58-59
The conventional multivariate flood loss estimation models often employ decision tree-based approaches to assess the role of different variables in influencing flood loss → Multivariate synthetic models also exist

Line 72
The objective of this study is to build a novel Flood Loss Estimation MOdel affected by flash floods (FLEMO*flash*) → check grammar

Line 254
The FLEMO*flash* model with the best performance, identified in Fig 3 → Which one is it? i.e., To which combinations of predictors, bins and neighbours correspond?

Line 256
C-GUI → Do authors mean C-BUI?

Line 256-257
*"For households (P:BUI and P:CON), the losses are significantly underestimated by the SDF-P"* → I cannot appreciate that

Line 276 -278
*"The CPT suggests that low water depths 275 (< 0.28 m) are most likely associated with low loss (< 0.05), while high water depths (> 0.15m) with high loss (> 0.24)"* → I would replace 0.05 with 0.17 and 0.15 with 1.5

Line 284-288
*"The CPT clearly indicates that contamination significantly amplifies the likelihood of experiencing higher loss (Fig 5). Specifically, when there is no contamination (class 0), the probability of experiencing loss is low (< 0.01). Conversely, if there is high contamination (class 4), the probability of experiencing loss is high (> 0.24), reflecting the impact of oils, chemicals, and sewage entering the building (Kreibich et al., 2005; Laudan et al., 2020)"* → it seems numbers are incorrect, please check or explain better

Line 292 → which is Figure 5e? see comment above

---

## Author Comment (AC1)

**Manuscript ID:** egusphere2025-1512

**Title: "FLEMO$_{flash}$ - Flood Loss Estimation MOdels for companies and households affected by flash floods"**

**Authors:** Apoorva Singh[1,2#], Ravi Kumar Guntu[1#*], Nivedita Sairam[1], Kasra Rafiezadeh Shahi[1], Anna Buch[3], Melanie Fischer[1], Chandrika Thulaseedharan Dhanya[2], and Heidi Kreibich[1]

We would like to express our sincere gratitude to the Editor for keeping the discussion open upon request and to the Reviewers for recognizing the significance of our work. We are especially thankful for their constructive comments and valuable suggestions, which we have carefully addressed in the revised version. The comments were found to be very helpful in enhancing the clarity and overall quality of the manuscript.

**Reviewer #1:**

This paper is in the context of flash floods, loss estimation models, and flood preparedness. The paper introduces the FLEMO$_{flash}$ model, using data from past German flash floods; methodologically, it combines machine learning and Bayesian networks to estimate probabilistic losses and their uncertainties. In terms of topics, the paper is relevant for and aligned with NHESS. The paper is well-written and -organised. Comments are mostly minor (even typos).

The authors would like to thank the reviewer for acknowledging significance and for providing us with the valuable feedback. The comments were found to be very helpful in improving the quality of the manuscript and will be acknowledged. We have responded (in black) to each comment (in blue). All references cited in our responses are listed at the end of this letter.

The only major comment is about preparedness. From the paper, I do not understand what is meant by preparedness, and in specific what 'high' and 'low' preparedness mean.

We thank the reviewer for this question. We have added the following text in Section 3.3.1 (P15/L323-330) to address the missing clarification.

*"Through feature selection and Bayesian Networks we identified emergency measures success ($ms$) and knowledge about emergency action ($ke$) for companies and private households respectively, as the significant variables (see Tables S2 and S3 for details on the questions and responses). Building on this, we conceptualised preparedness using these variables and categorized it into low, medium, and high levels. For companies, high preparedness was defined as having undertaken emergency measures that were perceived to be mostly or completely effective ($ms = 3$) and low preparedness ($ms = 1$) reflected low perceived effectiveness of such measures. For private households, high preparedness was defined as having a clear understanding of emergency actions based on official warnings ($ke \geq 5$), and low preparedness reflected limited to no understanding of what to do ($ke \leq 2$)."*

What are the assumptions behind 'preparedness'? e.g. that people with more knowledge of risk will act in a certain way (which way?)? At page 14, it is said: '…doesn't knew what to do'. For high preparedness, what people know about what to do?

Residents with high levels of preparedness are more likely to take effective emergency measures, thereby reducing the severity of flood loss. Despite its importance, the way preparedness is conceptualized in this study has certain limitations. Specifically, the variable does not capture which exact actions respondents undertook. Therefore, it would be misleading to speculate particular actions directly resulted in reduced losses. While the specific actions likely varied across respondents, empirical evidence indicates that having clear knowledge of emergency action generally contributes to better preparedness, consistent with previous findings.

We will mention this limitation in the revised manuscript in P16/L361-367 as follows:

*"Residents with high levels of preparedness are more likely to take effective emergency measures, thereby reducing the severity of flood loss (Kreibich et al., 2005; Sairam et al., 2019). Despite its importance, the way preparedness is conceptualized in this study has certain limitations. Specifically, the variable does not capture which exact actions respondents undertook. Therefore, it would be misleading to speculate particular actions directly resulted in reduced losses. While the specific actions likely varied across respondents, empirical evidence indicates that having clear knowledge of emergency action generally contributes to better preparedness, consistent with previous findings (Kreibich et al., 2021)."*

*The model seems suited to derive the predictive density of losses, however I have doubt about the effect of preparedness. I would be very cautious to include this part in the paper.*

Thank you for this helpful feedback. We have revised the manuscript to clarify how predictive densities are summarized. The following text has been added in P15/L330–339:

*"While preparedness has been extensively studied in the context of fluvial or riverine floods (Lüdtke et al., 2019; Schoppa et al., 2020; Wagenaar et al., 2018), its role in flash floods has not yet been systematically investigated. To address this gap, we applied the FLEMOflash model to derive predictive densities of rloss. Results were summarized using the median and associated uncertainty (25th and 75th percentiles) for selected combinations of hazard, exposure, and vulnerability conditions, rather than displaying the full predictive densities (Figure 6). For clarity of interpretation, Figure A1 illustrates step by step how predictive densities are derived from the prior and posterior distributions using kernel density estimation based on 1,000 resampled values, while Figure A2 provides an overview of the posterior and predictive densities across varying levels of measure success (preparedness) under same conditions of water depth and number of employees."*

[Figure]

**Figure A1: Visualizations of the prior, posterior, and predictive distributions of rloss (a) Empirical kernel density estimate of the prior rloss based on collected data (b) Prior distribution of rloss represented as bin weights (inverse frequency) across discretized intervals (c) Posterior distribution of rloss conditioned on wd ∈ [1.85, 2.40) and emp ≥ 38 (d) Predictive distribution of rloss generated by resampling 1000 values using the prior bin weights and the posterior probabilities. The solid vertical line indicates the median (50th percentile), while the dotted vertical lines represent the 25th and 75th percentiles, representing the predictive uncertainty. Shaded area highlight the interquartile range.**

[Figure]

**Figure A2: (a) Posterior distribution and (b) predictive density of relative loss (rloss) under condition of water depth (wd) ∈ [1.85, 2.40) and number of employees (emp) ≥ 38. (c–h) Posterior and predictive distributions of rloss for varying levels of measure success (ms): Subplots c, e, g present posterior distributions of rloss under three ms conditions — not at all, only limited, and yes – completely/most part — with wd ∈ [1.85, 2.40) and emp ≥ 38. Subplots d, f, h shows corresponding predictive densities, estimated using kernel density estimation from resampled values (n = 1000). In each density plot, the solid vertical line marks the median (50th percentile), while dotted vertical lines indicate the 25th and 75th percentiles, with shaded regions representing the uncertainty. The sequence from top to bottom illustrates increasing levels of preparedness.**

A secondary comment is that I would add some background about the previous /traditional version of FLEMO (e.g. https://www.gfz.de/en/section/hydrology/ projects/4-flood-loss-model-flemo-for-residential-and-commercial-sectors); there is none at the moment I think.

Thank you for the helpful suggestion. In the revised manuscript (P2-3/L54-75), we have incorporated additional background.

*"Traditionally, flood loss estimation relied on univariate stage-damage functions (SDF) (Middelmann-Fernandes, 2010). To improve the description of complex damage processes, the Flood Loss Estimation MOdel (FLEMOps) for the private sector, was developed as rule-based, multivariate, deterministic model (Thieken et al., 2008). Merz et al. (2013) and Sieg et al. (2017) introduced decision tree-based damage models that explicitly quantify uncertainty associated with both data variability and model structure uncertainty through bootstrap aggregation. Subsequently, Bayesian Networks were used (BN-FLEMO), enabling the modelling of complex flood loss processes through conditional probability relationships (Lüdtke et al., 2019; Schoppa et al., 2020; Schröter et al., 2014; Vogel et al., 2018).*

*In parallel, various machine learning approaches have also been developed for flood loss estimation, including neural networks (Salas et al., 2023), random forests (Ghaedi et al., 2022), Bayesian regression (Mohor et al., 2021). Among these, Bayesian networks are particularly advantageous due to their probabilistic representation of conditional dependencies among multiple variables, handle missing data, and model transferability (Schröter et al., 2014). Bayesian models enhance the understanding of flood loss dynamics by quantifying uncertainty and offering probabilistic estimates. For instance, Wagenaar et al. (2018) developed a regional and temporal transferable BN-FLEMO for microscale residential applications, which was later upscaled to mesoscale by Lüdtke*

*et al. (2019). In addition to the FLEMO typology, various synthetic, multivariate, rule-based flood loss models have been proposed for fluvial flood contexts (Amadio et al., 2019; Dottori et al., 2016; Nofal et al., 2020; Sairam et al., 2020).*

*However, all these loss models were developed to simulate damage processes during fluvial floods. In this study, we present the first probabilistic flash flood loss model – Flood Loss Estimation Model affected by flash floods (FLEMO$_{flash}$) using a BN-based approach and gain new knowledge about flash flood damage processes based on the conditional probabilities among multiple influencing variables. The study identifies the important variables and underlying processes that govern the flash flood losses. Additionally, we examine the predictive performance of FLEMO$_{flash}$ model and compare it with conventional SDF models. Finally, we illustrate the effect of preparedness in controlling the extent of loss reduction"*

Specific comments (P for page, L for line):

Valid for all direct citations: coma is not needed before the year, e.g. Smith et al. (2000) - and not Smith et al., (2000)

Thank you for pointing this out. We have corrected it.

Valid for the whole paper: equation factors, such as rloss, need to be in italic in the main text of the manuscript

Corrected.

Valid for the whole paper: do not use contracted forms like 'doesn't'. L223: The direction of the arrow represents an association between two variables but doesn't necessarily represent causality.

We have corrected it in the revised manuscript.

P2L42: double parenthesis in the citation

Corrected.

P2L50: double space before 'significant'

Corrected.

P4L101: double space before 'The percentage'?

Corrected.

P7L146, P11L236: 'This' what? Add a noun, specify

Thank you for pointing out this lack of clarity. The revised text now reads as follows:

[revised manuscript text omitted]

---

## Author Comment (AC2)

**Manuscript ID:** egusphere2025-1512

**Title: "FLEMO_flash - Flood Loss Estimation MOdels for companies and households affected by flash floods"**

**Authors:** Apoorva Singh[1,2#] , Ravi Kumar Guntu[1#*] , Nivedita Sairam[1] , Kasra Rafiezadeh Shahi[1] , Anna Buch[3] , Melanie Fischer[1] , Chandrika Thulaseedharan Dhanya[2] , and Heidi Kreibich[1]

We would like to express our sincere gratitude to the Editor for keeping the discussion open upon request and to the Reviewers for recognizing the significance of our work. We are especially thankful for their constructive comments and valuable suggestions, which we have carefully addressed in the revised version. The comments were found to be very helpful in enhancing the clarity and overall quality of the manuscript.

**Reviewer #2:**

The paper introduces FLEMOflash, a novel multivariate probabilistic Flood Loss Estimation Model tailored for flash floods. The model builds on survey data collected after flash flood events in 2002, 2016, and 2021 in Germany, encompassing both affected companies and households. FLEMOflash employs a data-driven feature selection approach alongside Bayesian networks to derive probabilistic loss estimates. The topic clearly falls within the scope of the journal, and the manuscript is generally well written and well organised. However, I have concerns regarding some of the underlying assumptions of the model, which, in turn, raise doubts about its validity for reliably estimating flash flood damage. I believe the authors should provide a more robust justification for their hypotheses to strengthen the credibility and robustness of their results. Below, I first present general concerns, followed by more specific comments.

The authors would like to thank the reviewer for acknowledging significance and for providing us with the valuable feedback. The comments were found to be very helpful in improving the quality of the manuscript and will be acknowledged in the revised manuscript. We have responded (in black) to each comment (in blue). All references cited in our responses are listed at the end of this letter.

**General concerns**

1) My first concern relates to the criteria used for identifying flash flood events (and then data to be implemented to derive the model). Specifically, I find the use of average slope as a proxy problematic. While slope may influence local flow velocity, it does not adequately capture the main defining characteristic of flash floods — their rapid onset and short lead times. This concern is further supported by the reported warning lead times in Tables 1 and 2, which range from 0 to 240 hours and 0 to 168 hours, respectively. These values appear inconsistent with typical flash flood dynamics, where lead times are often just a few hours. Additionally, the use of a low-resolution DEM may not provide the accuracy needed to derive reliable slope estimates at the point observation scale.

Have the authors considered using the concentration time of the river basin where the observations are located as a more physically meaningful proxy for flash flood potential? This could provide a better indication of response time and be more consistent with established hydrological understanding of flash flood processes.

We thank the reviewer for this important and constructive comment. The flood loss models presented in this study are based on empirical, microscale data collected from individual private households and companies. To identify flash flood samples, we applied a spatially informed terrain analysis. For this purpose, 14 reference municipalities with documented flash flood occurrences or described as particularly susceptible to flash floods were selected. An overview of these municipalities is provided in Table A1 (to be included in the revised manuscript).

**Table A1: Overview of 14 municipalities affected by past flash flood events.**

| Name of Municipality | Latitude (N) | Longitude (E) | Reference (including research papers, official reports, municipal flash flood maps, media coverage of past events) |
|---|---|---|---|
| Triftern | 48.3957 | 13.0060 | (LfU, 2017), Thieken et al. (2022) |
| Simbach am Inn | 48.2869 | 13.0113 | Hübl et al., (2017), (LfU, 2017), Thieken et al. (2022) |
| Obernzenn | 49.4492 | 10.4886 | (LfU, 2017), Thieken et al. (2022) |
| Künzelsau | 49.2802 | 09.7378 | Mühr et al. (2016), Thieken et al. (2022) |
| Julbach | 48.2547 | 12.9313 | (LfU, 2017), Thieken et al. (2022) |
| Forchtenberg | 49.2799 | 09.5149 | Mühr et al. (2016), Thieken et al. (2022) |
| Flachslanden | 49.4081 | 10.5205 | (LfU, 2017), Thieken et al. (2022) |

| Braunsbach | 49.2007 | 09.7873 | (Bronstert et al., 2018), Thieken et al. (2022) |
|---|---|---|---|
| Ansbach | 49.2888 | 10.5553 | (LfU, 2017), Thieken et al. (2022) |
| Stadtallendorf | 50.8308 | 09.02447 | AVOSS Test Municipality (https://www.avoss.uni-freiburg.de/testgebiete). Past event (https://www.feuerwehr-wetter.de/informationen/buergerinformationen/starkregen.html) |
| Grafschaft | 50.5752 | 07.0852 | AVOSS Test Municipality (https://www.avoss.uni-freiburg.de/testgebiete). Past event( https://hochwasser-grafschaft.de/?p=936) |
| Herrstein | 49.7845 | 07.3461 | AVOSS Test Municipality (https://www.avoss.uni-freiburg.de/testgebiete). Past event (https://fachtagung-funke.de/wp-content/uploads/2024/06/6_Fuhr_Eisatzbericht-Herrstein_2018.pdf) |
| Otting | 48.8801 | 10.7978 | AVOSS Test Municipality (https://www.avoss.uni-freiburg.de/testgebiete). Past event (https://www1.wdr.de/nachrichten/westfalen-lippe/aufraeumarbeiten-starkregen-ottfingen-100.html) |
| Emmendingen | 48.1225 | 07.8623 | AVOSS Test Municipality (https://www.avoss.uni-freiburg.de/testgebiete). Municipal flash flood maps (https://www.emmendingen.de/leben-umwelt/vorsorge-krise/starkregen) |

We agree with the reviewer that slope alone does not fully capture the characterization of flash floods. Other metrics, such as river basin concentration time, may indeed provide a more process-based representation of flash flood potential. Nevertheless, we used slope alone as a pragmatic solution to balance two competing needs: maintaining physical relevance in identifying flash flood–prone companies and households, and retaining a sufficient number of data points for robust model development.

We have added the following text in the manuscript (P4/L99–103):

*"Other metrics, such as river basin concentration time, may indeed provide a more process-based characterization of flash flood potential. Nevertheless, we used slope alone as a pragmatic solution that balances two competing needs: maintaining physical relevance in identifying flash flood prone companies and households, and retaining a sufficient number of data points for robust model development."*

P17/L373-377: *"While FLEMO$_{flash}$ already provides a robust tool to support risk analyses, and impact-based forecasting, future developments could further strengthen its applicability by integrating complementary hydrological indicators (e.g., basin concentration time), incorporating building-level susceptibility factors (e.g., construction materials, structural condition, floor count), and expanding the empirical database by including high loss observations and more diverse geographic regions."*

To assess the influence of DEM-granularity on our calculations, we compared the analysis results using the 90 m resolution DEM (SRTM GL3) to those acquired when using the 30 m resolution SRTM GL1 (see below Table, provided here for response only). We found that slope angles between the two medium-resolution DEMs generally increase with DEM-resolution, a relationship that is discussed in more detail by several studies (Chang and Tsai, 1991; Grohmann, 2015; Wu et al., 2008). However, it is arguable if these differences in calculated slope angles around the reference municipalities translate to significant differences in the selection of survey data points.

**Table**: Summary statistics (mean, median, minimum, and maximum) of terrain slope (in degrees) derived from SRTM GL3 (90 m resolution) and SRTM GL1 (30 m resolution) for the selected municipalities.

| Municipality | SRTM GL3 (90 m resolution) | | | | SRTM GL1 (30 m resolution) | | | |
|---|---|---|---|---|---|---|---|---|
| | mean | median | min | max | mean | median | min | max |
| Triftern | 1.75 | 1.49 | 0.00 | 6.06 | 5.37 | 4.30 | 0.00 | 38.78 |
| Simbach am Inn | 1.89 | 1.34 | 0.00 | 9.98 | 4.75 | 3.20 | 0.00 | 45.47 |
| Obernzenn | 1.68 | 1.41 | 0.01 | 7.22 | 4.69 | 3.04 | 0.00 | 37.71 |
| Künzelsau | 2.34 | 1.52 | 0.00 | 12.86 | 6.95 | 4.62 | 0.00 | 48.14 |
| Julbach | 1.62 | 1.21 | 0.00 | 9.98 | 4.97 | 3.45 | 0.00 | 48.70 |
| Forchtenberg | 2.40 | 1.92 | 0.00 | 10.74 | 7.21 | 5.16 | 0.00 | 54.09 |

| | | | | | | | | |
|---|---|---|---|---|---|---|---|---|
| Flachslanden | 1.84 | 1.70 | 0.01 | 6.80 | 5.42 | 3.84 | 0.00 | 37.81 |
| Braunsbach | 2.35 | 1.31 | 0.00 | 12.86 | 6.18 | 3.54 | 0.00 | 48.41 |
| Ansbach | 1.38 | 1.19 | 0.01 | 5.61 | 4.59 | 3.20 | 0.00 | 37.08 |
| Stadtallendorf | 1.72 | 1.55 | 0.00 | 10.81 | 4.45 | 3.30 | 0.00 | 43.48 |
| Grafschaft | 2.49 | 1.87 | 0.01 | 16.54 | 6.47 | 4.05 | 0.00 | 57.05 |
| Herrstein | 3.49 | 3.20 | 0.04 | 12.86 | 9.32 | 7.34 | 0.00 | 60.95 |
| Otting | 1.54 | 1.41 | 0.02 | 5.89 | 5.34 | 4.17 | 0.00 | 43.45 |
| Emmendingen | 2.42 | 1.59 | 0.00 | 13.96 | 7.79 | 4.66 | 0.00 | 59.19 |

Regarding the reported warning lead times, we agree that the values presented in Tables 1 and 2 appear long compared to typical flash flood dynamics. This discrepancy arises because, the variable "warning lead time" includes both flash flood warnings and heavy rainfall warnings. The latter are often issued days in advance by meteorological services, which explains the broader range (0–240 and 0–168 hours) seen in Tables 1 and 2. To clarify this, the revised manuscript now includes an overview of all variables for companies and private households (Tables S1 and S2), including the corresponding survey questions and responses.

2) My second concern relates to the set of explanatory variables used in the model. One of the primary damage mechanisms in flash flood events is structural damage, which is strongly influenced by the physical vulnerability of affected buildings. However, the model does not appear to include variables that capture this aspect, such as construction material, number of floors, or level of maintenance — all of which significantly affect a building's susceptibility to structural damage. While I understand that the set of variables was likely constrained by the information collected through the survey, I would like to know whether the authors considered integrating ancillary data to address these critical gaps. For example, building-level data from national censuses or geoportals could provide valuable proxies for physical vulnerability. Inclusion of such information could improve the explanatory power and practical relevance of the model, particularly in contexts where decisions rely on nuanced understanding of asset-specific vulnerabilities.

Thank you for your valuable and constructive suggestion. We fully recognize the importance of incorporating variables that directly reflect the physical vulnerability of buildings, as these factors significantly influence structural damage during flash flood events. Our current dataset already includes some relevant vulnerability-related variables (e.g., building area, size of premises, presence of a basement, and spatial situation), but it does not contain detailed information on construction materials or number of floors. In this study, our aim was to advance the understanding of processes and develop models based strictly on the available empirical survey data.

We also appreciate the suggestion of integrating ancillary data sources (e.g., open-source geoportals). While such data may indeed provide valuable proxies for building vulnerability, ensuring consistent integration across all surveyed municipalities was beyond the scope of the present study. Nevertheless, this represents a promising avenue for future research and model enhancement.

We have added the following text in the manuscript:

P17/L373-377: *"While FLEMO$_{flash}$ already provides a robust tool to support risk analyses, and impact-based forecasting, future developments could further strengthen its applicability by integrating complementary hydrological indicators (e.g., basin concentration time), incorporating building-level susceptibility factors (e.g., construction materials, structural condition, floor count), and expanding the empirical database by including high loss observations and more diverse geographic regions."*

3) A third concern regards obtained results, especially in terms of damage mechanisms. I would have expected to observe a significant influence of flow velocity or, at least, of the hydrodynamic force associated with the flow but this is not the case.

We thank the reviewer for this constructive comment. In the current study, we aimed to represent hydrodynamic forces through two variables: velocity and human stability. While the velocity variable reflects a subjective but

direct estimation of the local strength of the water flow by the interviewed people, the human stability variable captures the perceived difficulty of standing in floodwaters, thereby integrating both water depth and flow velocity. As shown in Figures 1(d–e), human stability emerges as the second most influential factor affecting loss in the case of private households, indicating that the combined effect of water depth and velocity is important for the model. We will include this explanation in Section 3.1 of the revised manuscript. Additionally, the revised manuscript now includes an overview of all variables for companies and private households (Tables S1 and S2), including the corresponding survey questions and responses.

P8/L197-203: *"Although flow velocity has been identified as a significant contributor to flash flood losses (Kreibich and Dimitrova, 2010), it does not appear among the most significant factors in the current study. In our analysis, we represent hydrodynamic forces using two variables: velocity (v) and human stability (hs). While velocity provides a subjective yet direct measure of local strength of flow current, human stability reflects on the perceived difficulty of standing in flood waters. As shown in Figure 1(d-e), human stability emerges as the second most influential factor affecting loss in private households, indicating that the combined effect of depth and velocity play a crucial role for the flash flood model."*

4) All the concerns mentioned above converge in the results obtained, particularly in the relative loss estimates provided by the model. These estimates range between 0.2 and 0.5, even for high water depths (around or above 2 meters). Such values are comparable to those typically produced by models for riverine floods (see, e.g., FLEMOps), which raises doubts about the model's ability to capture the distinctively more destructive nature of flash floods.

We agree that relative loss values between 0.2 and 0.5 may appear low compared to expectations for flash floods. However, two key factors explain this pattern.

First, our dataset contains a greater number of observations with lower reported damages compared to high-damage cases, resulting in a skewed distribution. This imbalance limits the model's ability to generalize accurately at the upper end of the water depth range. Please refer to Figure A1, which illustrates this distribution. Similar limitations have been reported in the literature; for example, Schoppa et al. (2020) observed greater prediction uncertainty for higher water depths due to data sparsity.

[Figure]

**Figure A1: Visualizations of the prior, posterior, and predictive distributions of rloss (a) Empirical kernel density estimate of the prior rloss based on collected data (b) Prior distribution of rloss represented as bin weights (inverse frequency) across discretized intervals (c) Posterior distribution of rloss conditioned on wd ∈ [1.85, 2.40) and emp ≥ 38 (d) Predictive distribution of rloss generated by resampling 1000 values using the prior bin weights and the posterior probabilities. The solid vertical line indicates the median (50th percentile), while the dotted vertical lines represent the 25th and 75th percentiles, representing the predictive uncertainty. Shaded area highlights the interquartile range.**

Second, our analysis (see Figure A2) shows that even under condition of higher water depth and high exposure (e.g., many employees), the level of preparedness, particularly the perceived success of emergency measures undertaken, plays a substantial role in reducing losses. Specifically, relative loss is significantly lower when respondents reported that the measures taken were either completely successful or protected the most critical parts of the property.

[Figure]

**Figure A2: (a) Posterior distribution and (b) predictive density of relative loss (rloss) under condition of water depth (wd) ∈ [1.85, 2.40) and number of employees (emp) ≥ 38. (c–h) Posterior and predictive distributions of rloss for varying levels of measure success (ms): Subplots c, e, g present posterior distributions of rloss under three ms conditions — not at all, only limited, and yes – completely/most part — with wd ∈ [1.85, 2.40) and emp ≥ 38. Subplots d, f, h shows corresponding predictive densities, estimated using kernel density estimation from resampled values (n = 1000). In each density plot, the solid vertical line marks the median (50th percentile), while dotted vertical lines indicate the 25th and 75th percentiles, with shaded regions representing the uncertainty. The sequence from top to bottom illustrates increasing levels of preparedness.**

While the resulting loss estimates may initially appear to underestimate the destructive nature of flash floods, they instead reflect the complex interplay between hazard intensity, exposure, and vulnerability. Nonetheless, we agree that increasing the number of data points representing extreme hazard scenarios and improving the representation of structural vulnerability (e.g., building materials, number of floors) would enhance the model's capacity to capture the full spectrum of flash flood impacts. We have emphasized these aspects as important directions for future study as follows (P17/L373-377):

*"While FLEMO_flash already provides a robust tool to support risk analyses, and impact-based forecasting, future developments could further strengthen its applicability by integrating complementary hydrological indicators (e.g., basin concentration time), incorporating building-level susceptibility factors (e.g., construction materials, structural condition, floor count), and expanding the empirical database by including high loss observations and more diverse geographic regions."*

**Specific Comments:**

Table 1 and Table 2 → the meaning of some variables is not clear. For instance, does emergency plan refer to the existence of a municipal emergency plan or a company emergency plan? Which is the meaning of the precaution indicator? Which are the emergency measures considered? I suggest including an explanatory table in the supplementary material

Thank you for highlighting this lack of clarity regarding the interpretation of variables in Tables 1 and 2. In the revised manuscript, we added overview of all variables for companies and private households (Tables S1 and S2), including the corresponding survey questions and responses.

**Table S1:** Overview of the company variables, including abbreviations, full variable names, survey questions, response options, coding, and index construction.

| Predictors | | Survey question | Response |
|---|---|---|---|
| $wd$ | Water depth | At maximum water level, how high was the water above the Earth's surface on your company premises in cm? | Continuous variable |
| $d$ | Inundation duration | For how many hours did water remain on the company premises? | Continuous variable |
| $v$ | Velocity indicator | How strong was the water current in the immediate vicinity of your company? | • 1 – Calm/slowly flowing
• 2
• 3
• 4
• 5
• 6 – Wild/violent current

Recoded categories (used in the analysis):

1. Low flow (original categories 1–2)
2. Moderate flow (original categories 3–4)
3. Torrential flow (original categories 5–6) |
| $con$ | Contamination | Did contamination from the following substances entered your company during the flood event? | *Response (with multiple options possible):*

• Oil/Gasoline
• Chemicals
• Sewage
• No contamination

Recoded categories (used in the analysis):

0. No contamination
1. Sewage or Chemicals only
2. Oil/Gasoline only
3. Oil/Gasoline + Sewage, or Oil/Gasoline + Chemicals
4. Oil/Gasoline + Chemicals + Sewage |
| $wt$ | Warning lead time | How many hours before the arrival of the flash flood or heavy rainfall did the warning reach your company? | • Number of hours
• No warning received |
| $ws$ | Early warning source | From which source did your company receive the flood warning? | *Response (with multiple options possible):*

• Loudspeaker announcements
• App or SMS
• Telephone call
• Radio report |

| | | | |
|---|---|---|---|
| | | | • TV report
• Newspaper report
• Social media
• Own research
• Own observation
• No warning

Recoded categories (used in the analysis):
0. No warning
1. Own research
2. Contacts (employees, acquaintances, other companies, phone calls)
3. Media (radio, TV, newspaper, online, social media)
4. Official authorities (direct official warning, apps/SMS, civil protection, loudspeaker announcements, regional services) |
| *ew* | Early warning received | Did your company receive an early warning of the flood event? | 0. No
1. Yes |
| *me* | Emergency measures undertaken | Were measures to reduce damage undertaken in your company before or during the flood event? | 0. No
1. Yes |
| *ep* | Emergency plan | At the time of the flood event, did your company have an emergency or flood protection plan? | 0. No
1. Yes |
| *kh* | Knowledge about hazard | Had this site already been flooded before?
Were you aware that your company is located in a flood-prone area? | 0. No
1. Yes |
| *ms* | Emergency measures success | Were measures to reduce damage undertaken in your company before or during the flood event?
How effective were these mitigation measures? | • No measure undertaken
• Not effective at all
• Only partly effective
• Mostly effective
• Completely effective

Recoded categories (used in the analysis):
0. No measure undertaken
1. Completely ineffective,
2. Partly effective,
3. Mostly/ completely effective |
| *fe* | Flood experience | Q1: Had this company site already been flooded before the event? If yes, how many times? | Number of previous floods:
0. Never
1. Once
2. Twice
3. Three times
4. Four times
5. More than four times |

| | | | |
|---|---|---|---|
| | | Q2: When was the company site last affected by a flood prior to the event? (Year) | Time elapsed since the last flood:
1. 25 years ago
2. 10–25 years ago
3. 5–10 years ago
4. 2–5 years ago
5. 0–2 years ago |
| *ss* (Spatial situation row region) | | Flood experience was calculated from the number of previous floods (Q1) and the time elapsed since the last flood (Q2). | • If only one value (Q1 or Q2) was available, that value was used.
• If both values were available, the flood experience score was calculated as the mean of the two. |
| *pr* | Precaution indicator | *Measures included*

V1. Company insured against flood damages.
V2. Heating system adjusted (converted or flood-protected).
V3. Emergency plan in place.
V4. Frequency of emergency drills conducted before the flood.
V5. Tanks, silos, or storage facilities securely anchored.
V6. Stationary or mobile water barriers installed.
V7. Sensitive equipment relocated to higher floors.
V8. Water-hazardous substances relocated to higher floors.
V9. Use of flood-prone areas adapted to risk.
V10. Air conditioning/ventilation system flood-proofed.
V11. Building flood safety improved (e.g., sealing basements, strengthening stability). | Conversion:
• Each measure was coded as 1 if implemented prior to the flood, 0 otherwise.
• For drills, any positive frequency ($\geq 1$ per year) was coded as 1, absence as 0.

Weighting scheme:
• Low impact / basic preparedness (weight = 1): V1 to V4
• Medium impact / protective but limited scope (weight = 5): V5 to V8
• High impact / comprehensive protection (weight = 10): V9 to V11

Calculation of weighted score ($p$):
$p = v1 + v2 + v3 + v4 + (5 \times (v5 + v6 + v7 + v8)) + (10 \times (v9 + v10 + v11))$
Precaution Indicator ($pr$):

0. No precautionary measures
1. Medium precaution ($p: 1 - 5$)
2. Very good precaution ($p \geq 6$) |
| *in* | Insurance | Is the company insured against flood damages before the flood event? | 0. No
1. Yes |
| *sec* | Sector | Which sector does your company belong to? | 1. Agriculture
2. Manufacturing
3. Trade
4. Finance
5. Services |
| *ss* | Spatial situation | Which description best fits the spatial situation of this flood-affected company site? | 1. Business premises with several buildings belonging to the company
2. Entire building fully used by the company |

| | | | 3. One or more floors in a building otherwise used for non-business purposes
4. Less than one floor in a building otherwise used for non-business purposes |
|---|---|---|---|
| *own* | Ownership | Are the buildings or rooms owned by the company or rented? | 1. Owned
2. Rented
3. Partly owned / partly rented |
| *emp* | Number of employees | How many people were employed in the previous month? | Continuous variable |
| *sp* | Size premise | How large is the property on which your company is located? | Continuous variable |

**Table S2:** Overview of the private household variables, including abbreviations, full variable names, survey questions, response options, coding, and index construction.

| | Predictors | Survey question | Response |
|---|---|---|---|
| *wd* | Water depth | At the maximum water level: How high did the water stand approximately outside the building? | Continuous variable |
| *d* | Inundation duration | For how many hours did the water remain inside the building in total? | Continuous variable |
| *v* | Velocity scaled | How strong was the water current in the immediate vicinity of your house? | 0. No flow
1. Calm flowing
2. .
3. .
4. .
5. .
6. Torrential flow |
| *hs* | Human stability | Do you think an average man could have stood upright in the flood near your house? | 1. Person can stand effortlessly in calm water,
2. Should make effort to stand,
3. Person would have been swept away,
4. Too deep to stand |
| *con* | Contamination | Was your affected property contaminated by the following substances? | *Response (with multiple options possible):*

• Oil/Gasoline
• Chemicals
• Sewage
• No contamination

Recoded categories (used in the analysis):

0. No contamination
1. Sewage or Chemicals only
2. Oil/Gasoline only
3. Oil/Gasoline + Sewage, or Oil/Gasoline + Chemicals
4. Oil/Gasoline + Chemicals + Sewage |

| | | | |
|---|---|---|---|
| *ew* | Early warning received | How did you become aware that the flood danger was becoming acute for you? | 0. No warning received
1. Warning received |
| *wt* | Warning lead time | How many hours before the onset of flooding did the warning reach you, or did you yourself become aware of the danger? | Continuous variable |
| *ws* | Warning source | How did you become aware that the flood danger would become acute for you? | 0. No warning received
1. Own observation
2. Contacts
3. Media
4. Official warning through authorities |
| *ke* | Knowledge about emergency action | Before the flood danger became acute: Did you know how you and your household could protect yourselves against flooding from heavy rainfall? | 1. It was completely unclear to me
2. .
3. .
4. .
5. .
6. It was completely clear to me |
| *me* | Emergency measures undertaken | Did you – or someone else – take measures to reduce damages in your house? | 0. No
1. Yes |
| *mu* | Number of emergency measures undertaken | Did you – or someone else – take measures to reduce damages in your house? | (Nominal: 0 = No, 1 = Yes)

• Secured documents and valuables
• Moved/secured furniture and movable items
• Secured oil tanks or other containers
• Pumped out or scooped water
• Brought animals to safety
• Moved vehicles to flood-safe place
• Protected building against water intrusion
• Redirected water flow on property
• Received help from outside
• Unplugged electronic devices
• Dismantled fixed electrical installations
• Shut off gas/electricity manually
• Gas/electricity shut off centrally by authorities
• No measure taken

Score = documents + furniture + oil + pump + pets + car + building + redirect + help + unplugged + dismantled + $gas_{self}$ + $gas_{authority}$

• Minimum = 0 (No measure undertaken)
• Maximum = 13 (All measures undertaken) |
| *fe* | Flood experience | Q1: How often were you personally affected by heavy rainfall or floods before the event? | Number of previous floods:
0. Never
1. Once
2. Twice
3. Three times
4. Four times
5. More than four times |

| | | Q2: When was the last time you were affected by a flood or heavy rainfall-related inundation? (Year) | Time elapsed since the last flood:
1. 25 years ago
2. 10–25 years ago
3. 5–10 years ago
4. 2–5 years ago
5. 0–2 years ago |
|---|---|---|---|
| | | Flood experience was calculated from the number of previous floods (Q1) and the time elapsed since the last flood (Q2). | • If only one value (Q1 or Q2) was available, that value was used.
• If both values were available, the flood experience score was calculated as the mean of the two. |
| $pw$ | Precaution indicator | *Measures included*

V1. I find out how to protect my house/flat against flooding.
V2. I take out insurance against flood damage
V3. I participate in neighborhood flood assistance.
V4. I use flood-prone floors in a low-value way (adapted use).
V5. I avoid valuable permanent fittings in flood-prone storeys and use water-resistant/renewable materials (adapted furniture).
V6. I relocate the heating system and/or electrical supply to higher floors.
V7. I change the heating system or flood-protect the oil tank.
V8. I improve the safety of the building (e.g. seal basements)
V9. I install stationary or mobile water barriers.
V10. I prepare for emergencies (e.g. water pumps, generator). | Conversion:
• Each measure was coded as 1 if implemented prior to the flood, 0 otherwise.

Weighting scheme:
• Low impact (weight = 1): V1 to V4
• Medium impact (weight = 5): V6 to V10
• High impact (weight = 10): V4, V5
Calculation of weighted score ($p$):
$$p = v1 + v2 + v3 + v4 + (5 \times (v6 + v7 + v8 + v9 + v10)) + (10 \times (v4 + v5))$$
Precaution Indicator ($pw$):
   0. No/Low precaution ($p < 7$)
   1. Medium precaution ($7 \leq p < 25$)
   2. Very good precaution ($p \geq 25$) |
| $fa$ | Building footprint area | What is your estimate of the building's floor area? | Continuous variable |
| $b$ | Basement | Does the building have a full or partial basement? | 0. No basement
1. Partial basement
2. Full basement |
| $per$ | Household size | How many people live permanently in your household, including yourself and all children? | Continuous variable |
| $chi$ | Number of children | How many children under 14 years of age live in your household? | Continuous variable |

| eld | Number of elders | How many people in your household are older than 65? | Continuous variable |
|---|---|---|---|
| inc | Monthly net income in classes | What is the approximate total monthly net income of your household in euros? | 1. < 500 €
2. 500-1000
3. 1001-1500
4. 1501-2000
5. 2001-3000
6. > 3000 € |
| socp | Socioeconomic status according to Plapp, (2003) | What is your highest educational qualification? | 1. No school degree
2. Lower secondary
3. Secondary school
4. Vocational or technical qualification
5. Higher education |

Living condition: Derived from ownership structure and building type

Ownership structure:

1. Tenant
2. Apartment owner
3. House owner

Building type:

1. Single-family house
2. Multi-family house
3. Semi-detached house

| Ownership | Building type | Living condition |
|---|---|---|
| 1 (Tenant) | 2 (multiple) | 1 |
| | 1 (single) | 2 |
| | 3 (semi-detached) | 2 |
| 2 (Apartment owner) | | 3 |
| 3 (House owner) | | 4 |

What is the total usable living area of the house (all floors together, but without the basement)?

$$living\ space = \frac{usable\ area}{household\ size}$$

1. Less than 25%
2. 25% to < 50%
3. 50% to < 75%
4. 75% or more

$$Socp = Education + Living\ condition + Living\ space$$

- Minimum value: 3 (if all indicators are at their lowest)
- Maximum value: 13 (if all indicators are at their highest)

Line 102-104: "*To maximise the amount of training data for model building, we employed the nearest neighbour technique to impute the missing data. We tested a range of k-neighbours for our datasets (k =1,3,5,7,9) and selected the value with best performance*" → while this could be a good option for spatially correlated variables such as velocity and warning lead time (after verifying that the distance between points is limited), it may lead to misleading assumptions for other missing variables. For example, variables such as in, sp, and sec (for companies) or ke, fa, and b (for buildings) are not necessarily spatially correlated. It would be helpful if the authors could provide a more thorough discussion on this point, particularly addressing the potential limitations and implications of their imputation strategy for these types of variables.

We thank the reviewer for this valuable comment. In our dataset, missing values occurred because not all respondents answered every survey question. To avoid significant data loss, we employed the k-nearest neighbours (kNN) imputation method. We emphasize that the imputation was based on similarity in feature space, rather than on spatial proximity. The kNN algorithm identified the most similar observations across all available variables to impute missing entries, regardless of their geographic locations. We acknowledge that this assumption may be more suitable for certain variables than for others that may not exhibit strong correlation with other features.

While kNN imputation is effective for preserving data quantity and minimizing loss, it introduces certain assumptions and limitations. Primarily, it assumes that missing values can be reasonably predicted based on similarity to other observations in the dataset. Additionally, imputation can reduce the natural variability of the data, potentially leading to additional uncertainty in the modelling results. Despite these limitations, our analysis showed that the box plots and distributions remained stable after imputation (not shown for brevity). Nevertheless, we advise interpreting the results involving imputed variables with caution and recommend further validation using complete datasets in future studies.

In the revised manuscript we added the following text to Section 3.2.2 (P11/L250–256):

*"The k-nearest neighbours (kNN) method of imputation assumes that the missing values can be inferred based on similarity of feature space. This may not hold equally well across variables, particularly for those with weak correlation to other features. To evaluate the robustness of imputation process, we compared the distribution of variables before and after imputation and found them to be largely consistent (not shown for brevity). Nevertheless, the imputation process may still introduce uncertainty or reduce natural variability in the data. Future studies could benefit from sensitivity testing using alternative imputation techniques and explore models that explicitly incorporate imputation uncertainty."*

Section 3.1

The meaning of two CPTs in the table (d-con, wd-hs) should also be discussed. Moreover, I think this section should be expanded discussing results for all damage components (i.e. companies BUI, EQU, GNS and household CON), even without reporting all the CPTs.

We thank the reviewer for this valuable suggestion. We have provided a more comprehensive explanation of all the damage components as follows (P13-14/L288-313):

*"The loss processes described by FLEMO$_{flash}$ is illustrated using the predictive density of predicted losses under scenarios of hazard, exposure and vulnerability. For brevity, this section primarily focusses on the FLEMOflash model for household buildings (Fig 5), with a similar interpretation extended to other asset types (Fig S3-S6). The nodes of the model comprise of water depth, human stability, inundation duration, contamination, knowledge about emergency action, and relative losses, each with 7, 4, 7, 5, 6, and 8 classes, respectively. The Conditional Probability Table (CPT) was populated with joint probabilities to find the predictive density of loss given the condition of other nodes.*

*The conditional probability of* `rloss` *based only on water depth indicates a monotonic relationship. Shallow inundations are associated with very low losses, while deeper water substantially increases the probability of severe losses (Fig 5e). The highest probabilities are concentrated along the diagonal, confirming this trend. For instance, depths <0.28 m are most likely associated with very low losses (<0.017), whereas depths ≥2.3 m are strongly associated with high losses (> 0.42). Similar patterns of increasing loss probability with greater water depth are observed across all asset types (Fig. S3–S6). Water depth also influences human stability: while shallow flooding results in low instability, extreme depths markedly increase the probability of instability (0.54) (Fig. 5b, Fig. S6a).*

*Contamination emerges as another important driver of losses. In uncontaminated conditions (class 0), the probability of very low losses (<0.01) is high (0.82). Conversely, under severe contamination (class 4), the probability of very high losses (>0.427) increases to 0.30 (Fig. 5c), reflecting the destructive impact of oils, chemicals, and sewage entering buildings (Kreibich et al., 2005; Laudan et al., 2020). Households exposed to inundation lasting [13–50) hours showed a high probability of experiencing moderate contamination levels (classes 1–2). Knowledge about emergency action shows a strong mitigating effect. The CPT (Fig. 5d) demonstrates that households with low awareness (Ke ≤ 2) face a high probability of severe losses, whereas*

*households with very good knowledge (Ke ≥ 5) display a substantially higher probability of reduced losses. Comparable findings are observed for household contents (Fig. S6c). This agrees with Kreibich et al. (2021), who also reported that clear awareness of emergency actions substantially reduces damages. Importantly, socioeconomic status indirectly shapes vulnerability, as higher-income groups are more likely to report very clear knowledge of emergency actions after receiving warnings (Fig. S6b).*

*For companies (Figs. S3–S5), the CPT results reveal consistent patterns across buildings, equipment, and goods & stock. Smaller companies (with fewer employees or smaller premises) show higher probabilities of severe losses, whereas larger firms and premises are more strongly associated with lower loss outcomes (Figs. S3b, S4d, S5c). Across all asset types, the success of emergency measures emerges as a dominant factor, as unsuccessful measures are strongly associated with a high probability of severe losses (Figs. S3d, S4b, S5a). Contamination further amplifies losses, with severe categories linked to markedly higher probabilities of loss. Together, these results emphasize that hazard intensity (water depth, velocity, contamination), exposure (number of employees, size premises) and vulnerability factors (effectiveness of emergency measures) interactively determine relative losses for companies."*

Figure 5 → I think that results explanation will be supported if each CPT is identified with a letter

In the revised manuscript, we labeled each subplot in Figure 5 and Figures S3-S6.

[Figure]

**Figure 3. Conditional probability table (CPT) of the Bayesian network for the residential building. Each heatmap illustrates the conditional probabilities of a child node given its parent node. Parent node states are shown along the x-axis, and child node states along the y-axis. Darker shades of blue indicate higher probability values. Numerical values are displayed in each cell, with an accompanying colorbar showing the probability scale. (a) Inundation duration given contamination (b) Water depth given human stability (c) Contamination given relative loss (d) Knowledge about emergency action given relative loss (e) Relative loss given water depth.**

Line 289- 296 *"The integration of knowledge about emergency action into the FLEMO_{flash} model alongside water depth and contamination provides a comprehensive understanding of how preparedness can mitigate loss during flash floods. Knowledge about emergency action is categorized into six classes, ranging from 1 (low knowledge) to 6 (high knowledge). The CPT clearly illustrates that a high level of emergency action knowledge can significantly reduce loss (Fig 5e). Specifically, when households doesn't knew what to do (1), there is a high likelihood of incurring higher loss. Conversely, when households with good preparedness (> 4), the incurred loss significantly decreases. Residents with high levels of preparedness are more likely to take effective emergency measures, thereby reducing the severity of flood loss"* → Knowing what to do does not necessarily imply that individuals will take action. Do the authors have any insight into why this variable appears to be significant in the model, potentially even more so than the actual implementation of protective measures (me, mu)?

We thank the reviewer for raising this important question. Residents with high levels of preparedness are more likely to take effective emergency measures, thereby reducing the severity of flood loss. Despite its importance, the way preparedness is conceptualized in this study has certain limitations. Specifically, the variable does not capture which exact actions respondents undertook. Therefore, it would be misleading to speculate particular actions directly resulted in reduced losses. While the specific actions likely varied across respondents, empirical evidence indicates that having clear knowledge of emergency action generally contributes to better preparedness, consistent with previous findings.

We will mention this limitation in the revised manuscript in P16/L361-367 as follows:

*"Residents with high levels of preparedness are more likely to take effective emergency measures, thereby reducing the severity of flood loss (Kreibich et al., 2005; Sairam et al., 2019). Despite its importance, the way preparedness is conceptualized in this study has certain limitations. Specifically, the variable does not capture which exact actions respondents undertook. Therefore, it would be misleading to speculate particular actions directly resulted in reduced losses. While the specific actions likely varied across respondents, empirical evidence indicates that having clear knowledge of emergency action generally contributes to better preparedness, consistent with previous findings (Kreibich et al., 2021)."*

**Minor comments**

Line 58-59: The conventional multivariate flood loss estimation models often employ decision tree-based approaches to assess the role of different variables in influencing flood loss → Multivariate synthetic models also exist

Thank you for the suggestion. We will revise the introduction to include mention of existing multivariate synthetic models as follows (P2-3/L54-75):

*"Traditionally, flood loss estimation relied on univariate stage-damage functions (SDF) (Middelmann-Fernandes, 2010). To improve the description of complex damage processes, the Flood Loss Estimation MOdel (FLEMOps) for the private sector, was developed as rule-based, multivariate, deterministic model (Thieken et al., 2008). Merz et al. (2013) and Sieg et al. (2017) introduced decision tree-based damage models that explicitly quantify uncertainty associated with both data variability and model structure uncertainty through bootstrap aggregation. Subsequently, Bayesian Networks were used (BN-FLEMO), enabling the modelling of complex flood loss processes through conditional probability relationships (Lüdtke et al., 2019; Schoppa et al., 2020; Schröter et al., 2014; Vogel et al., 2018).*

*In parallel, various machine learning approaches have also been developed for flood loss estimation, including neural networks (Salas et al., 2023), random forests (Ghaedi et al., 2022), Bayesian regression (Mohor et al., 2021). Among these, Bayesian networks are particularly advantageous due to their probabilistic representation of conditional dependencies among multiple variables, handle missing data, and model transferability (Schröter et al., 2014). Bayesian models enhance the understanding of flood loss dynamics by quantifying uncertainty and offering probabilistic estimates. For instance, Wagenaar et al. (2018) developed a regional and temporal transferable BN-FLEMO for microscale residential applications, which was later upscaled to mesoscale by Lüdtke et al. (2019). In addition to the FLEMO typology, various synthetic, multivariate, rule-based flood loss models have been proposed for fluvial flood contexts (Amadio et al., 2019; Dottori et al., 2016; Nofal et al., 2020; Sairam et al., 2020).*

*However, all these loss models were developed to simulate damage processes during fluvial floods. In this study, we present the first probabilistic flash flood loss model – Flood Loss Estimation Model affected by flash floods (FLEMOflash) using a BN-based approach and gain new knowledge about flash flood damage processes based on the conditional probabilities among multiple influencing variables. The study identifies the important variables and underlying processes that govern the flash flood losses. Additionally, we examine the predictive performance of FLEMOflash model and compare it with conventional SDF models. Finally, we illustrate the effect of preparedness in controlling the extent of loss reduction"*

 The objective of this study is to build a novel Flood Loss Estimation MOdel affected by flash floods (FLEMO$_{flash}$) → check grammar

The above line has been removed in the revised manuscript and has been replaced with:

P3/L70-73: *"In this study, we present the first probabilistic flash flood loss model – Flood Loss Estimation Model affected by flash floods (FLEMO$_{flash}$) using a BN-based approach and gain new knowledge about flash flood damage processes based on the conditional probabilities among multiple influencing variables."*

 The FLEMO$_{flash}$ model with the best performance, identified in Fig 3 → Which one is it? i.e., To which combinations of predictors, bins and neighbours correspond?

We have revised the figure caption and mention the best-performing configurations as follows:

*Figure 4. Model sensitivity of FLEMO$_{flash}$ to the number of predictors (f1–f5), bins (b3–b8), and number of neighbours used for data imputation (k1–k9), evaluated using mean absolute error (MAE), continuous ranked probability score (CRPS), and mean bias error (MBE) for the five asset types (x-axis). Each boxplot summarizes 100 repetitions of fivefold cross-validation (companies) and tenfold cross-validation (households) with randomized data partitioning. Best-performing configurations were identified through a sequential tuning process: first selecting the number of predictors based on the first panel, then optimizing bin count in the second panel with predictors fixed, and finally selecting the number of neighbours in the third panel with both previous parameters fixed. Best-performing configurations are: Companies – Buildings (C:BUI) f5, b6, k7; Companies – Equipment (C:EQU) f5, b6, k5; Companies – Goods and Stock (C:GNS) f5, b6, k9; Private Households – Buildings (P:BUI) f5, b8, k1; Private Households – Contents (P:CON) f5, b8, k3.*

 C-GUI → Do authors mean C-BUI?

Corrected. We meant C:GNS.

*"For households (P:BUI and P:CON), the losses are significantly underestimated by the SDF-P"* → I cannot appreciate that

We thank the reviewer for this observation. The corresponding statement has been removed in the revised manuscript.

 *"The CPT suggests that low water depths 275 (< 0.28 m) are most likely associated with low loss (< 0.05), while high water depths (> 0.15m) with high loss (> 0.24)"* → I would replace 0.05 with 0.17 and 0.15 with 1.5

Thank you for pointing out this typo-error. We have revised the statement as follows:

[revised manuscript text omitted]

---

## Author Response (AR2)

The authors would like to thank the Editor and reviewers for the positive and encouraging feedback. The additional comments were found to be very helpful in improving the quality of the manuscript and will be acknowledged. We have responded (in black) to each comment (in blue). Please note that the page and line numbers cited in our responses refer to the clean version of the manuscript.

**Reviewer #1:**

As a reviewer of the previous version, I do appreciate the improvement made in the paper. However, I remain dubious about the 'preparedness' part. It still reads/looks as a (quick) addition, and not a key, well-thought contribution in the paper. In fact:

- the Introduction does not mention preparedness at all (and then no background/SOTA on it);

- the Methodology has not description about the methods for the preparedness analysis, except Tables 1/2 (which are not explained in the details of preparedness).

The preparedness contribution is squeezed into Sec. 3.3.1 that mixes methodology and results. As a result, the quality of this contribution is very low.

I would remove it, or I would revisit the paper (from the structure to content), in order to reflect how this aspect is embedded in the study.

The authors would like to thank the reviewer for acknowledging the improvements done in the revised manuscript. We understand that the presentation of "preparedness" compromises with the continuity and clarity of the manuscript. As suggested by the reviewer we have removed the term "preparedness" unless in places where it means a general term rather than a conceptualized indicator. We have removed the section titled "Effect of preparedness". As understanding the role of adaptation strategies such as emergency measures success ($ms$) and knowledge about emergency action ($ke$) is an important finding of the study, we have re-written the section with more clarity. We hope that this re-written section provides more clarity and suits the flow of the manuscript.

P15/ L 321- 351:

[revised manuscript text omitted]

**Reviewer #2:**

I would like to thank the authors for their thorough and timely response to my comments, as well as for the revisions made to the manuscript, which in my view provide an appropriate justification for the modelling choices adopted and the results obtained.

Nevertheless, I still have some reservations regarding the data underpinning the analysis and their representativeness with respect to the flash flood phenomena. While the use of the slope criterion allows for the inclusion of a larger number of data points, it may not necessarily ensure the quality or suitability of the dataset for this specific context. In addition, I remain uncertain about the inclusion of events related to heavy rainfall. The relatively low damage values estimated by the model (even in the absence of mitigation measures) seem to support these concerns. This aspect, however, does not preclude the publication of the article, since, as already mentioned, the manuscript clearly explains the modelling choices adopted.

We sincerely thank the reviewer for the constructive feedback and positive evaluation of our revised manuscript. We appreciate the reviewer for the insightful comments.

The slope-based approach was adopted as a simple and pragmatic criterion for point-based samples, allowing the inclusion of a broader empirical dataset while maintaining consistency across study sites. We fully acknowledge, however, that this criterion may select cases where the flash flood situation was not particularly severe. As highlighted in the conclusions, future developments should complement this approach with more physically based hydrological modelling and additional indicators. Specifically, we noted

P16/L365-369:

"*While FLEMO$_{flash}$ already provides a tool to support risk analyses, and impact-based forecasting, future developments could further strengthen its applicability by integrating complementary hydrological indicators (e.g., basin concentration time), incorporating building-level susceptibility factors (e.g., construction materials, structural condition, floor count), and expanding the empirical database by including high loss observations and more diverse geographic regions.*"

In the methods section (P4/L100-101), we have already added "*Other metrics, such as river basin concentration time, may indeed provide a more process-based characterization of flash flood potential.*" to highlight the limitation of considering only slope criterion.

We agree that we have used a rather broad definition of flash floods. However, the three flood events used in this study are already mentioned in section 2.1, of which one is a heavy precipitation event. We did not use rainfall as a criterion to distinguish flash floods, and only heavy precipitation events which qualified the slope criteria were used in this study. We have added a disclaimer to this effect in the methods section

P4/L103-104:

"*The cases with longer warning lead times in the sample are likely to be due to warnings of high precipitation than to flood-specific warnings.*"

Nevertheless, I would suggest that the authors adopt a more cautious tone when discussing:

-(i) the model's ability to identify "the important variables and underlying processes that govern flash flood losses". In fact, the model essentially indicates that damages depend on flood intensity, exposure, and preparedness — aspects that are already well-established in the literature.

We agree with the reviewer in using a more cautious tone when discussing the models' ability to identify important variables. It is true that the model essentially indicates that damages depend on flood intensity, exposure, and preparedness which have been well-established in the existing literature. However, using a machine learning ensemble approach helps us do quantitative comparison of factors that influence loss, and select the most significant variables from a list of 20 variables considered in this study. This final selected list of variables is used for Bayesian Networks, which in turn help draw probabilistic dependencies between different variables.

In Section 3.1, we clearly highlight and acknowledge the role of existing studies stating that our results are in line with the previous findings. We have rephrased sentences that mention the "identification" of flash flood loss drivers as a novelty and highlight how the quantitative machine learning approach helps identify which of the variables are "more" important.

We have made the following changes in response to the suggestions made by the reviewer:

P3/L74-76:

"*We use machine learning based feature importance to select the most important variables from our dataset. The performance of FLEMO$_{flash}$ model is compared against conventional SDF models. Finally, we illustrate the loss processes with the CPT and Markov blanket in controlling the extent of loss reduction.*"

P6/ L120-121

"*To derive the most significant drivers of flash flood losses from our list of variables, this study adopts a data-driven feature selection approach to the empirical data.*"

P8/L 195-197:

"*By quantitatively comparing the varied drivers of flash flood losses, these results also emphasize the importance of multivariable loss estimation models that capture the interplay across these drivers and their influence on losses.*"

- (ii) the predictive capacity of the model (for example, in the abstract and conclusions), given that the model relies on a limited number of variables and is based on data that are difficult to estimate in a predictive phase at the individual-item level (e.g., preparedness, number of employees, income, contamination, inundation duration).

As mentioned in Section 2.4.2, the model predictive capacity presented in this study pertains to an evaluation of model's performance with respect to available data and established metrics (e.g., MAE, CRPS, MBE). We acknowledge that these metrics reflect outcome constrained by available data and model choices.

As a word of caution, we have added the following part to the conclusion of the manuscript:

P16/L369-371:

"*It is important to note that the model's performance and predictive capacity, as presented are specific to the empirical dataset and survey variables available for FLEMO$_{flash}$, and the results should be interpreted within the context and limitations of the underlying data.*"

---

## Author Response (AR3)

**Editor:**

Thank you very much for the submission of your manuscript on 'FLEMOflash - Flood Loss Estimation MOdels for companies and households affected by flash floods' and for answering the reviewer's comments on it.

I am happy to accept it for publication in NHESS. Congratulations!

Some technical edits should be considered:

- l 76: CPT is used for the first time here. Please add 'conditional probability table (CPT)'

- Labels and legend of figures 3, 6, A1, A2 are hardly readable.

- The predictive density plots in Fig. 6 are lacking a scale.

The authors would like to sincerely thank the Editor for accepting the manuscript and for providing additional technical suggestions, which were very helpful in further improving the quality of the manuscript. We have addressed all suggestions as detailed below:

- At line 76, the abbreviation CPT is now defined as conditional probability table (CPT) at its first occurrence.
- The font size of labels and legends in Figures 3, A1, and A2 has been increased to improve readability. In addition, high-quality versions of all figures have been provided to facilitate the production process.
- In Figure 6, redundant panels have been removed while preserving the original scientific content, and the missing scale has been added in the legend (see revised figure).

[Figure]

**Figure 6. FLEMOflash application for company buildings, private household buildings and contents, considering relative loss Markov blankets. The first row in each panel shows the probabilistic predictive density of relative loss on the interval [0,1] based on the specific scenarios of hazard and exposure combination. The second to fourth rows in each panel illustrate the changes in relative loss with different levels of emergency measures success (ms) and knowledge about emergency action (ke) for the given hazard and exposure combinations.**

On behalf of the authors, thank you very much once again for your time and support.

Warm regards,

Ravikumar Guntu